# High-resolution projections of ambient heat for major European cities using different heat metrics

Clemens Schwingshackl[1,2], Anne Sophie Daloz[2], Carley Iles[2], Kristin Aunan[2], and Jana Sillmann[3,2]

[1]Department of Geography, Ludwig-Maximilians-Universität München, Munich, Germany
[2]Center for International Climate Research (CICERO), Oslo, Norway
[3]Center for Earth System Research and Sustainability (CEN), University of Hamburg, Hamburg, Germany

*Correspondence to*: Clemens Schwingshackl (c.schwingshackl@lmu.de)

**Abstract.** Heat stress in cities is projected to strongly increase due to climate change. The associated health risks will be exacerbated by the high population density in cities and the urban heat island effect. However, impacts are still uncertain, which is among other factors due to the existence of multiple metrics for quantifying ambient heat and the typically rather coarse spatial resolution of climate models. Here we investigate projections of ambient heat for 36 major European cities based on a recently produced ensemble of regional climate model simulations for Europe (EURO-CORDEX) at 0.11° spatial resolution (~12.5 km). The 0.11° EURO-CORDEX ensemble provides the best spatial resolution currently available from an ensemble of climate model projections for the whole of Europe and makes it possible to analyse the risk of temperature extremes and heatwaves at the city-level. We focus on three temperature-based heat metrics – yearly maximum temperature, number of days with temperatures exceeding 30 °C, and Heat Wave Magnitude Index daily (HWMId) – to analyse projections of ambient heat at 3 °C warming in Europe compared to 1981-2010 based on climate data from the EURO-CORDEX ensemble. The results show that southern European cities will be particularly affected by high levels of ambient heat, but depending on the considered metric, cities in central, eastern, and northern Europe may also experience substantial increases in ambient heat. In several cities, projections of ambient heat vary considerably across the three heat metrics, indicating that estimates based on a single metric might underestimate the potential for adverse health effects due to heat stress. Nighttime ambient heat, quantified based on daily minimum temperatures, shows similar spatial patterns as daytime conditions, albeit with substantially higher HWMId values. The identified spatial patterns of ambient heat are generally consistent with results from global Earth system models, though with substantial differences for individual cities. Our results emphasise the value of high-resolution climate model simulations for analysing climate extremes at the city-level. At the same time, they highlight that improving the predominantly rather simple representations of urban areas in climate models would make their simulations even more valuable for planning adaptation measures in cities. Further, our results stress that using complementary metrics for projections of ambient heat gives important insights into the risk of future heat stress that might otherwise be missed.

## 1 Introduction

Global heat stress is projected to strongly increase in the future due to climate change (Gasparrini et al., 2017; Vargas Zeppetello et al., 2022; Zheng et al., 2021; Schwingshackl et al., 2021; Freychet et al., 2022), and already nowadays record-breaking high temperatures are observed more and more often around the world, such as in Canada in summer 2021 (White et al., 2023) or in China and Europe in summer 2023 (Zachariah et al., 2023). Heat stress can have severe implications for human health, the economy, and the society as a whole (e.g., McMichael et al., 2006; Gasparrini et al., 2015; Yang et al., 2021; Alizadeh et al., 2022; Orlov et al., 2021), as it can lead to decreased levels of comfort and reduced labour productivity (Orlov et al., 2021; García-León et al., 2021), enhanced socioeconomic inequalities (Alizadeh et al., 2022), and increased morbidity and mortality (Gasparrini et al., 2015). Moreover, as the health risk associated with heat stress is not uniform within the population, heatwaves and extreme temperatures pose a larger threat to those who are most vulnerable to elevated temperatures, particularly to children, older adults, and persons with pre-existing conditions (Lundgren et al., 2013).

Various metrics have been developed with the aim to capture the characteristics of heat extremes, including heatwaves, and their potential evolution in the future (e.g., Perkins and Alexander, 2013; Perkins, 2015; de Freitas and Grigorieva, 2017). Several of these indicators are based solely on temperature, while others consider additional factors, such as humidity, solar radiation, or wind speed to estimate heat exposure (de Freitas and Grigorieva, 2017). In the following, we focus on temperature-based metrics, given that many epidemiological studies found temperature to be the dominant factor for adverse health effects (Armstrong et al., 2019; Kent et al., 2014; Vaneckova et al., 2011). Future changes in heat and heat extremes are frequently quantified by the change in temperature (e.g., mean or maximum near-surface air temperature) between a historical reference period and future periods (Sillmann et al., 2013; IPCC, 2021; Coppola et al., 2021). Other studies used the number of days per year during which certain thresholds are exceeded (e.g., Casanueva et al., 2020; Schwingshackl et al., 2021; Zhao et al., 2015). Likewise, different metrics have been introduced to quantify heatwaves, often based on percentile-based thresholds (e.g., Fischer and Schär, 2010; Suarez-Gutierrez et al., 2020; Perkins-Kirkpatrick and Lewis, 2020). The Heat Wave Magnitude Index daily (HWMId, Russo et al., 2015) integrates both the magnitude and the length of a heatwave into a single metric to quantify the heatwave severity. HWMId was applied by several studies to analyse the future risk of heatwaves (e.g., Dosio et al., 2018; Russo et al., 2017; Forzieri et al., 2016; Zittis et al., 2021). Depending on the considered metric, the projected spatial patterns of ambient heat projections may vary considerably, highlighting that assessing the future risk from heat stress requires considering a portfolio of metrics.

The health risk from heat stress is not spatially homogeneous – neither globally nor within a country or a region – owing to several factors, including variations in local climate conditions, local climate feedbacks (e.g., due to albedo, soil moisture), or differences in the social environment (e.g., population density, socioeconomic conditions). Temperatures are often amplified in cities due to the predominance of impervious surfaces and the multitude of anthropogenic heat sources. The resulting urban heat island (UHI) effect leads to higher levels of ambient heat in cities compared to surrounding areas (e.g., Heaviside et al., 2017). In Europe, our region of study, about 75% of the population lives in urban areas (UN-Habitat, 2011) and the urban

population is projected to grow even further in the future along with an ageing trend (Smid et al., 2019). Larger metropolitan areas in Europe will become more vulnerable to extreme heat in the coming decades (Smid et al., 2019) and heat mortality in European cities is projected to significantly increase (Karwat and Franzke, 2021). Cities in Europe or elsewhere are thus becoming climate hotspots in terms of climate change (Zheng et al., 2021) but also for adaptation and innovation (IPCC, 2022) due to the need for adequate strategies to address climate change adaptation. Preventing adverse health outcomes from heat stress and designing appropriate and effective adaptation measures requires accurate projections and estimates of heatwaves and temperature extremes. Recently, climate model simulations have reached a spatial resolution high enough to provide such projections at the city-level.

Analyses of climate and climate change in cities face the challenge of delivering results on spatial resolutions that are high enough to be relevant for cities while robustly estimating the risk of extreme events. Urban canopy layer models, which can resolve cities at scales of ~100 m or even higher, can deliver great spatial details of cities (e.g., Masson et al., 2020), with the trade-off that often only a limited number of cities are examined (e.g., Goret et al., 2019; Krayenhoff et al., 2020). Analyses with urban canopy layer models coupled to climate models often rely on data from a single or a few climate models and are thus not able to adequately incorporate climate variability to robustly quantify the probability of extreme events. On the other hand, climate model simulations can be used to quantify climate variability and the risk of extreme events in multiple cities. Guerreiro et al. (2018) used simulations by general circulation models (GCMs) from the Climate Model Intercomparison Project phase 5 (CMIP5) to investigate heatwave projections in European cities. However, GCMs cannot fully depict local urban climate conditions as the spatial resolution of GCMs (~100 km) is much coarser than that of urban canopy layer models. To provide higher spatial resolution and to overcome some of the limitations of GCMs, dynamical downscaling by regional climate models is frequently applied. This approach has been used multiple times to investigate individual cities with a single model (e.g., Argueso et al., 2015; Chapman et al., 2019; Keat et al., 2021; Kusaka et al., 2012; Li and Bou-Zeid, 2013; Ramamurthy and Bou-Zeid, 2017; Wouters et al., 2017) but rarely for analysing climate conditions in a large number of cities and/or with an ensemble of models (e.g., Sharma et al., 2019; Smid et al., 2019; Junk et al., 2019). For Europe, an ensemble based on regional climate models (RCMs) from the European branch of the Coordinated Regional Downscaling Experiment (EURO-CORDEX; Jacob et al., 2013; Vautard et al., 2021) is available, providing simulations at a resolution of 0.11° (EUR-11, ~12.5 km), which is fine enough to analyse climate conditions in major European cities at the city-level as typically at least one model grid cell falls within the extent of each major European city. The EUR-11 simulations were evaluated by Coppola et al. (2021) and Vautard et al. (2021) who showed that the simulations reproduce well the observed spatial temperature distribution in Europe, despite a general cold bias of summer temperatures of around 1 °C to 2 °C compared to observation-based data from E-OBS (Cornes et al., 2018) in large parts of Europe. Hot biases of extreme temperatures (i.e., hottest five consecutive days) in mountainous regions are reduced in EURO-CORDEX compared to CMIP5, while a cold bias remains in central and northern Europe and a warm bias in southern Europe (Iles et al., 2020). Lin et al. (2022) evaluated the representation of HWMId in a subset of the EURO-CORDEX ensemble against reanalysis data, finding overall good agreement between both datasets and highlighting the added value of RCMs compared to the driving GCMs for representing small-scale features.

EURO-CORDEX simulations have been used to examine how temperatures and ambient heat are projected to increase in the
future throughout Europe (Vautard et al., 2013; Molina et al., 2020; Coppola et al., 2021) and for a small group of European
cities (Junk et al., 2019; Langendijk et al., 2019; Burgstall et al., 2021), showing that urban areas will be strongly affected by
rising temperatures. The different studies used varying sets of metrics, different model ensembles, and different selections of
cities. Smid et al. (2019) analysed HWMId projections for European capitals based on eight EURO-CORDEX models at 0.11°
resolution, focusing on the metropolitan areas around the capitals. They found highest HWMId increases in southern European
cities and, additionally, they highlight that exposure to heatwaves also strongly depends on population density. Junk et al.
(2019) analysed projections of several heatwave metrics defined by the Expert Team on Climate Change Detection and Indices
(ETCCDI) for London, Luxembourg, and Rome based on 11 EURO-CORDEX models at 0.11° resolution. The considered
heatwave metrics project strongest increases for Rome, except for the number of heatwaves per year, which the authors explain
by the increasing length of heatwaves, reducing their number. Using wet-bulb globe temperature (WBGT) as a heat metric,
Casanueva et al. (2020) analysed exceedances of WBGT thresholds above 26 °C and 28 °C in Europe based on an ensemble
of 39 EURO-CORDEX models (using simulations at both 0.11° and 0.44° resolution). Future exceedances of WBGT>28 °C
are projected to be highest in southern Europe, followed by central Europe, while exceedance rates are negligible in northern
Europe. Based on CMIP5 GCMs, Guerreiro et al. (2018) found that strongest increases in heatwave days are projected for
southern European cities along with substantial increases in coastal cities in northern Europe, while maximum temperatures
of heatwaves are projected to rise most strongly in central Europe.
Here we build on these studies and use simulations by 72 GCM-RCM model combinations of the 0.11° EURO-CORDEX
ensemble to assess projections of ambient heat for 36 major European cities. We focus on temperature and compare three
metrics: changes in yearly maximum near-surface air temperature, the number of days per year on which daily maximum near-
surface air temperature exceeds 30 °C, and HWMId. To evaluate potential differences in projections for daytime and nighttime
conditions, we additionally consider daily minimum near-surface air temperature. We first analyse how well the EURO-
CORDEX ensemble reproduces the measured temperature distributions in the selected cities compared to reanalysis and
observation-based data. Further, we quantify how ambient heat is projected to evolve in these cities under global warming
according to the three considered heat metrics. Finally, we evaluate how the choice of metrics affects projections of ambient
heat, which can give relevant insights for designing appropriate adaptation measures to counteract health risks from ambient
heat. A holistic analysis of the health risk from heat stress comprises the factors heat-related hazards, heat exposure, and
vulnerability to heat. We focus on the hazard from extreme heat by employing the three heat metrics, acknowledging that
exposure and vulnerability can also vary strongly across cities (Smid et al., 2019; Sera et al., 2019; Gasparrini et al., 2015).

## 2 Data and Methods

### 2.1 Data

### 2.1.1 Cities

We include 36 major European cities in our analysis. These comprise all European cities with a population of more than 1.2 million, and all European capitals with more than 500,000 inhabitants. We register the coordinates and elevation of each city, and whether it is located close to the sea (see Supplementary Table S1). A city is considered to be located close to the sea if it is directly adjacent to the sea. The complete list of cities and their geographic locations are indicated in Figure 1a.

### 2.1.2 Climate model data

The analysis is based on 72 GCM–RCM model chains from the EURO-CORDEX ensemble, which covers the European domain (Jacob et al., 2013, see Supplementary Table S2 for a detailed list of models). EURO-CORDEX simulations use two different spatial resolutions, 0.11° (EUR-11, ~12.5 km) and 0.44° (EUR-44, ~50 km). We only use data from the higher-resolution EUR-11 simulations, for which typically at least one grid cell falls within the extent of each major European city (Figure 1b). For our analysis, we use daily maximum near-surface air temperature (tasmax), daily minimum near-surface air temperature (tasmin), and monthly mean near-surface air temperature (tas), employing data from historical and RCP8.5 simulations for the period 1981-2100 (note that some model simulations only run until 2099 and one only until 2098). Near-surface air temperature refers to the temperature at 2 m height. For each city, we use data from the grid cell that is located closest to the centre of each city. The large ensemble of 72 GCM–RCM model combinations allows for a robust estimation of future ambient heat including the model structural uncertainty, which has been shown to be relevant for quantifying the risk of urban heatwaves (Zheng et al., 2021). To test the spatial robustness of our results, we additionally consider data from a box of 3x3 grid cells around the city centres. The representation of urban areas varies considerably across RCMs (Table 1). Some RCMs represent urban areas as rock surfaces, others assume reduced vegetation and adjusted surface parameters (such as albedo and roughness) for urban areas, and one RCM even includes a sophisticated urban model.

We further use simulations from the CMIP5 (24 models) and CMIP6 (24 models) ensembles (using one ensemble member per model) for comparison with the EURO-CORDEX simulations (see Supplementary Tables S3 and S4 for a detailed list of the considered CMIP5 and CMIP6 models and ensemble members). We employ data from historical and RCP8.5 simulations (SSP5-8.5 in case of CMIP6), analysing daily maximum near-surface air temperature (tasmax) and monthly mean near-surface air temperature (tas) for the same period (1981-2100) as for EURO-CORDEX. Analogous to EURO-CORDEX, we use the grid cell closest to the city centre for our analysis. To evaluate how the downscaling of GCMs by RCMs affects the results, we further consider the CMIP5 model set that is used to drive the 72 EURO-CORDEX RCMs. For this purpose, we create a GCM ensemble, which we denote as "EURO-CORDEX GCM ensemble", for which we consider each GCM member as many times as it is used as a driving GCM in the EURO-CORDEX ensemble. The EC-EARTH ensemble member r3i1p1 (used to drive

several EURO-CORDEX RCMs, see Supplementary Table S2) is not available via the Earth System Grid Federation (ESGF)
data portals and we thus substitute it by the EC-EARTH member r1i1p1 to create the EURO-CORDEX GCM ensemble.
The GCMs and RCMs used in this study differ in several aspects. Most importantly, the RCMs have a much higher spatial
resolution (~12.5 km) than the GCMs (~100 km), and orography and coastlines are thus represented much more accurately in
RCMs. GCMs and RCMs also differ in their projections of atmospheric aerosols over the European domain, with GCMs using
future scenarios with decreasing atmospheric aerosol concentrations while RCMs assume a constant atmospheric aerosol load
(Boé et al., 2020; Gutiérrez et al., 2020; Nabat et al., 2020). Additionally, unlike GCMs, several RCMs do not consider plant
physiological $CO_2$ effects, which might cause an underestimation of temperature extremes (Schwingshackl et al., 2019).

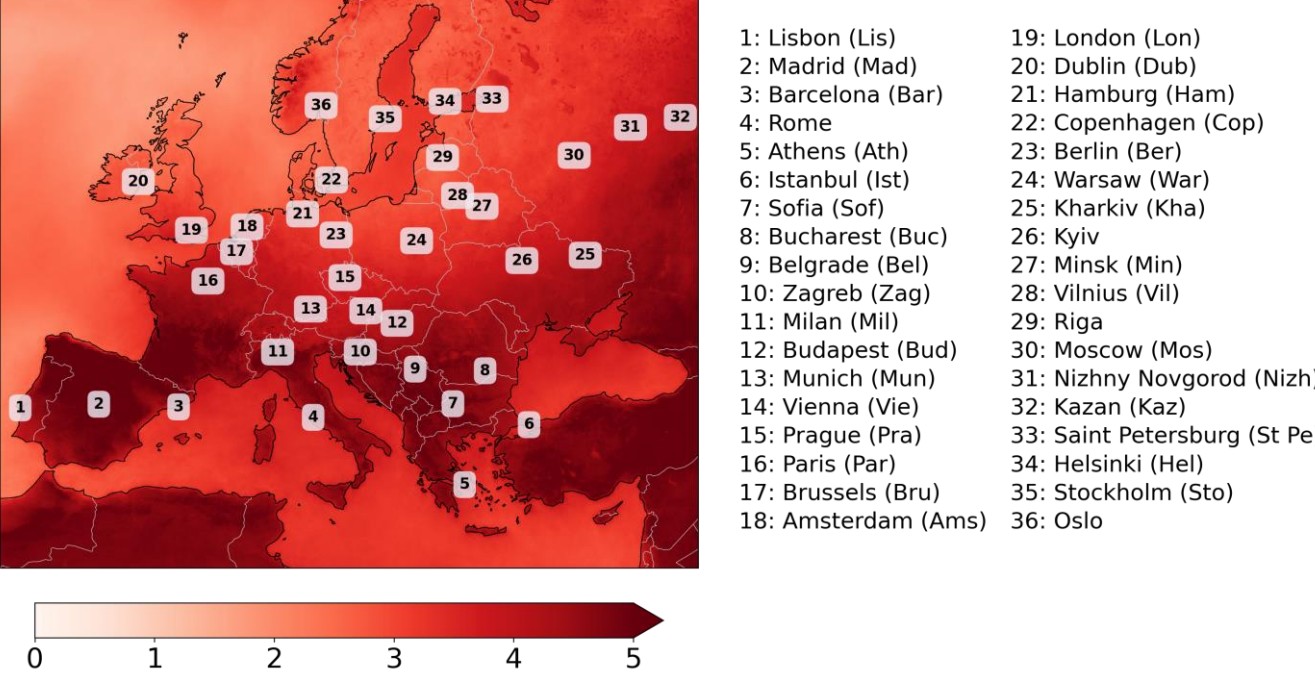

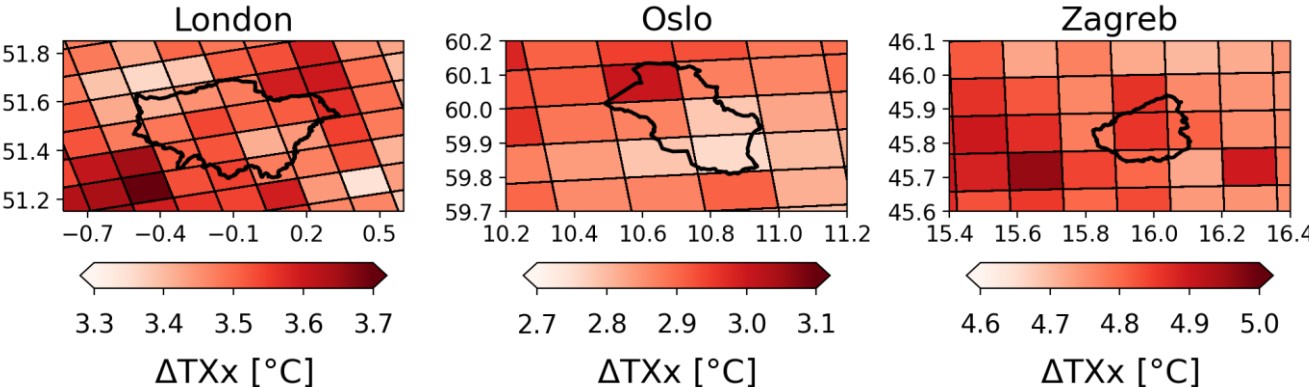

**Figure 1:** Overview of the cities investigated in this study and examples of the spatial resolution of EURO-CORDEX models. Top: Location of the cities with the background map showing the EURO-CORDEX multi-model median change of annual maximum near-surface air temperature (ΔTXx) at 3 °C European warming relative to 1981-2010 (see Section 2.2). Abbreviations in the list of cities indicate the abbreviated city names used in Figure 7. Bottom: Example of grid spacing used by the majority of EURO-CORDEX models compared to the extent of three cities with different sizes (black polygons).

73  **Table 1:** Representation of urban areas in the regional climate models of the 0.11° EURO-CORDEX ensemble (EUR-11).

| Institute | Model | Data source | Representation of urban areas | References |
|---|---|---|---|---|
| CLMcom | CCLM4-8-17 | Land-surface model TERRA | natural surfaces with an increased surface roughness length and a reduced vegetation cover | (Garbero et al., 2021; Doms et al., 2011) |
| CLMcom-ETH | COSMO-crCLIM-v1-1 | Land-surface model TERRA | natural surfaces with an increased surface roughness length and a reduced vegetation cover | (Garbero et al., 2021; Doms et al., 2011) |
| CNRM | ALADIN53 | ECOCLIMAP-II database | same as for rocks; no vegetation | (Daniel et al., 2018), pers. communication Samuel Somot (CNRM, 13/10/2023) |
| CNRM | ALADIN63 | ECOCLIMAP-II database | same as for rocks; no vegetation | (Daniel et al., 2018; Decharme et al., 2019) |
| DMI | HIRHAM5 | ECHAM5 | adjusted constant surface parameters; vegetation not mentioned | (Langendijk et al., 2019; Roeckner et al., 1996, 2003) |
| MPI-CSC | REMO2009 | Land Surface Parameter dataset of Hagemann (2002) | adjusted albedo and roughness length; no vegetation | (Jacob et al., 2012; Langendijk et al., 2019; Hagemann, 2002) |
| GERICS | REMO2015 | Land Surface Parameter dataset of Hagemann (2002) | adjusted albedo and roughness length; no vegetation | (Jacob et al., 2012; Remedio et al., 2019) |
| ICTP | RegCM4-6 | Land-surface model CLM4.5, which integrates the Community Land Model Urban (CLMU) | CLMU considers canyon geometry, pervious and impervious surfaces, roofs, and walls and distinguishes between four levels of urbanization; vegetation is considered as part of pervious surfaces | (Oleson and Feddema, 2020; Oleson et al., 2010, 2013) |
| IPSL | WRF381P | Standard canopy model from Unified Noah land-surface model (the urban canopy model implemented in WRF was not used for the EURO-CORDEX simulations) | bulk urban parameterization, increased surface roughness length; reduced vegetation cover | (Niu et al., 2011; Shen et al., 2022; Chen et al., 2011), pers. communication Linh Luu (University of Lincoln, 10/10/2023) |
| KNMI | RACMO22E | ECOCLIMAP version 1 | no specific urban parameterization but adjusted roughness length; vegetation not mentioned | (van Meijgaard et al., 2008), pers. communication Erik van Meijgaard (KNMI, 8/11/2023) |
| MOHC | HadREM3-GA7-05 | JULES Global Land 7.0 | urban canopy with thermal properties of concrete; adjusted roughness length and albedo; no vegetation | (Best et al., 2011; Walters et al., 2019) |
| SMHI | RCA4 | ECOCLIMAP version 1 | same as for rocks (urban areas not explicitly mentioned in documentation); no vegetation | (Samuelsson et al., 2015), pers. communication Patrick Samuelsson (SMHI, 27/10/23) |

74

### 2.1.3 Reference datasets

We evaluate the EURO-CORDEX simulations by comparing them against two gridded reference datasets (see Section 3.1): 1) the E-OBS gridded meteorological dataset, which provides gridded meteorological fields interpolated from weather station data at 0.1° resolution for Europe (Cornes et al., 2018) and 2) the global reanalysis ERA5-Land, which provides land variables including 2 m air temperature at a spatial resolution of about 9 km (Muñoz-Sabater et al., 2021). Additionally, we use data from single weather stations that lie within or close to the considered cities, using data from the Global Surface Summary of the Day (GSOD; Smith et al., 2011) and from the European Climate Assessment & Development (ECA&D; Klein Tank et al., 2002; Klok and Klein Tank, 2009). We only include data from weather stations with a data record length of at least 20 years. For all datasets, the evaluation is performed using daily maximum near-surface air temperature and daily minimum near-surface air temperature in the period 1981-2010. For ERA5-Land, daily maximum and daily minimum near-surface air temperatures are calculated as maximum and minimum of the hourly 2 m air temperature data. The land scheme of ERA5-Land does not specifically consider urban areas (ECMWF, 2018) and thus, specific climatic conditions in cities (such as the urban heat island effect, UHI) may not be fully represented. For cities, in which temperature data from weather stations within the city limits are assimilated in ERA5-Land or considered in E-OBS, UHI might, however, be partly included.

### 2.2 European mean warming

Regional temperatures and temperature extremes scale linearly with global mean surface air temperature (GSAT; Seneviratne et al., 2016; Wartenburger et al., 2017; Seneviratne and Hauser, 2020). Uncertainties connected to the underlying climate scenarios can thus be reduced if expressing future evolutions of regional temperatures as a function of changes in GSAT, usually calculated relative to pre-industrial (1850-1900) conditions. This approach of expressing climate change in terms of global warming levels instead of emission-driven or concentration-driven scenarios has been used by several recent studies (e.g., Schwingshackl et al., 2021; Freychet et al., 2022; Li et al., 2021) and was widely applied in the 6[th] Assessment Report of the Intergovernmental Panel on Climate Change (IPCC, 2021). While this approach works well on global scales, it cannot be applied directly to the regional climate model simulations of EURO-CORDEX, mainly due to two reasons. First, EURO-CORDEX simulations only start in 1950 (some models in 1970) and pre-industrial reference temperatures are therefore not available. We thus derive changes in mean temperatures relative to the period 1981-2010. Second, the EURO-CORDEX ensemble projects a lower rate of warming in Europe than the CMIP5 ensemble (Coppola et al., 2021). This discrepancy has been attributed to several reasons, such as differences in aerosol forcing (Boé et al., 2020; Gutiérrez et al., 2020; Nabat et al., 2020) or diverging trends in cloudiness (Bartók et al., 2017). To account for this discrepancy, we implement the scaling approach using European mean surface air temperature (ESAT) instead of GSAT based on temperature data from the EURO-CORDEX simulations. We calculate GSAT and ESAT from monthly mean temperature (tas), where ESAT is defined as the average temperature of a box spanning over Europe from 10° W to 35° E and from 30° N to 70° N.

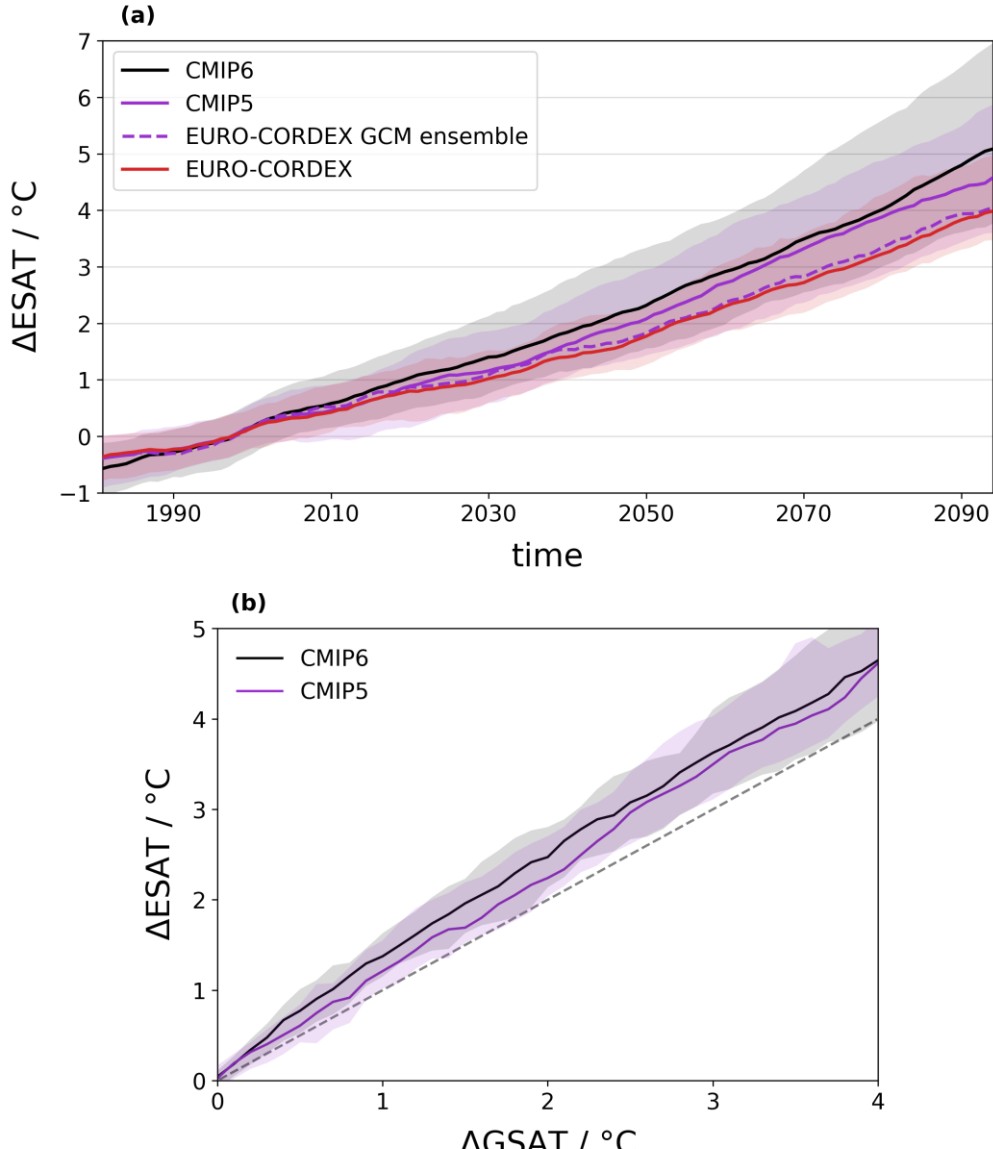

207

**Figure 2:** Warming in Europe in the RCP8.5 scenario (EURO-CORDEX, CMIP5) and SSP5-8.5 scenario (CMIP6) relative to
1981-2010. (a) Change in European mean surface air temperature (ESAT) as a function of time. The dashed purple line
indicates the EURO-CORDEX GCM ensemble (see Section 2.1.2 for more details). (b) Change in ESAT as a function of
change in global mean surface air temperature (GSAT) relative to the reference period 1981-2010. Solid lines in (a) and (b)
indicate the multi-model median and shading the range from 10th to 90th percentile across models. Data in (a) are smoothed
with a 10-year window and data in (b) are interpolated in 0.1 °C steps. The dashed grey line in (b) represents the identity line.

Comparing the warming projections in the CMIP5, CMIP6, and EURO-CORDEX ensembles (Figure 2a) confirms that the
CMIP5 and CMIP6 ensembles project a faster warming in Europe than the EURO-CORDEX ensemble. However, if
considering the EURO-CORDEX GCM ensemble (see Section 2.1.2), the resulting warming projections are very similar to
the projections of the EURO-CORDEX ensemble. This indicates a general agreement between the warming projections of
CMIP5 and EURO-CORDEX averaged over Europe and suggests that the difference in ESAT is mainly connected to the GCM
subset used to drive the EURO-CORDEX RCMs. As ESAT scales well with GSAT (Figure 2b), the warming can also be
directly related to changes in GSAT.
For consistency, we choose to stay within the EURO-CORDEX framework and express our results as a function of ESAT
instead of GSAT, based on temperature data from the EURO-CORDEX simulations. The results are shown for a European
warming of 3 °C relative to 1981-2010. This corresponds to a global warming of 2.5 °C in CMIP5 (2.4 °C to 2.7 °C;
interquartile range across models) and of 2.4 °C in CMIP6 (2.3 °C to 2.6 °C) relative to 1981-2010 and to a global warming
of around 3.1 °C in CMIP5 (3.0 °C in CMIP6) since pre-industrial conditions (1850-1900), which lies within the range of
global warming projections under current policies and actions (2.1 °C to 3.5 °C by 2100 based on the assessment by Climate
Action Tracker, https://climateactiontracker.org, last access 09 November 2023). For each GCM–RCM model chain of EURO-
CORDEX, we estimate the model-specific time when ESAT increases by 3 °C relative to 1981-2010 using a 20-year window
around the first year in which the 20-year average temperature exceeds 3 °C warming. The same approach is applied to CMIP5
and CMIP6 model data.
**2.3 Metrics for quantifying ambient heat**
Three heat metrics are used in this study to quantify how ambient heat will change in European cities under global warming.
The selected metrics were applied in various studies to investigate projections of ambient heat in Europe and globally (e.g.,
Casanueva et al., 2020; Lin et al., 2022; Coppola et al., 2021; Russo et al., 2015; Dosio et al., 2018). The first metric is the
change in yearly maximum temperature (TXx; based on daily maximum near-surface air temperature data) between the
reference period 1981-2010 and the (model-specific) time when European warming reaches 3 °C relative to 1981-2010. The
change in TXx indicates how strongly extreme temperatures increase due to climate change.
As a second metric we calculate the number of days per year on which daily maximum near-surface air temperature (TX)
exceeds 30 °C at the time when European warming reaches 3 °C. The threshold of 30 °C is a compromise of being high enough
to be relevant for southern European countries and low enough for northern European countries. While absolute thresholds
have been used in several scientific studies (e.g., Zhao et al., 2015; Schwingshackl et al., 2021; Casanueva et al., 2020), it
should be kept in mind that exceedances of absolute thresholds strongly depend on local climate conditions. To test the
sensitivity to the selected threshold level, we investigate how varying the threshold between 25 °C and 33 °C affects the
identified geographic patterns. Calculating exceedances of fixed thresholds based on climate model data usually requires bias
adjustment to correct for potential model biases (Maraun, 2016). However, we do not apply bias adjustment here due to the
lack of reliable reference data, given that urban areas are not specifically represented in the reference datasets ERA5-Land,

and E-OBS only implicitly includes information about urban areas to the extent weather stations are present within the city limits (which does not apply to all analysed cities, see Figure 3). Consequently, the urban heat island effect might be underrepresented in these datasets. Instead, we test the effect of a simple adjustment method that 1) adjusts the mean of the climate model data to ERA5-Land, and 2) adjusts the mean and variability to ERA5-Land (i.e., by applying a transformation to standard score). For this purpose, the mean and standard deviation of daily maximum and daily minimum near-surface air temperatures in summer (June, July, August) are calculated for each grid cell in a box of 5x5 grid cells around the centre of each city in the reference period 1981-2010. The resulting values are averaged over the 5x5 box and used for the simple adjustment method. The 5x5 box is used to represent the larger-scale climatological conditions within and around each city. The rationale is to reduce the statistical uncertainty by basing the adjustment on 25 grid cells instead of just one. The ERA5-Land data is bilinearly interpolated to the grid of each EURO-CORDEX model before calculating the mean and standard deviation. We use a Kolmogorov-Smirnow test to check whether the bias-adjusted heat metrics are statistically significantly different from the heat metrics calculated from the original data.

The third metric that we apply is the Heat Wave Magnitude Index daily (HWMId, Russo et al., 2015), which integrates both the length and the magnitude of a heatwave to calculate its overall strength. In the context of HWMId, heatwaves are defined as at least three consecutive days with daily maximum near-surface air temperatures above the 90th percentile of the daily maximum near-surface air temperature distribution of all days within a 31-day window in a pre-defined reference period (Russo et al., 2015). For each day in a heatwave, the HW magnitude ($HW_M$) is calculated by subtracting the 25th percentile of TXx ($TXx_{25p}$) in the reference period 1981-2010 from daily maximum near-surface air temperature (TX), normalised by the interquartile range of TXx in the reference period:

$$HW_M = \begin{cases} \frac{TX - TXx_{25p}}{TXx_{75p} - TXx_{25p}}, & \text{if } TX > TXx_{25p} \\ 0, & \text{otherwise} \end{cases} \qquad (1)$$

The sum over all daily HW magnitudes of a heatwave yields HWMId. By definition, HWMId takes into account the interannual temperature variability of each location. We calculate HWMId using daily maximum near-surface air temperature (denoted as HWMId-TX) for the time when European warming reaches 3 °C with 1981-2010 as the reference period. In each year, we identify the heatwave with the highest HWMId-TX and use it to calculate the 20-year average HWMId-TX.

To represent nighttime conditions, we further calculate the three different heat metrics based on daily minimum near-surface air temperature (TN), i.e., the yearly maximum of daily minimum near-surface air temperatures (TNx), the number of tropical nights (TN>20 °C), and HWMId based on daily minimum near-surface air temperature (HWMId-TN).

## 2.4 Statistical analysis

### 2.4.1 Spatial patterns of ambient heat

To analyse how a city's geographic location and local climate affect projections of ambient heat according to the three metrics, we estimate the contribution of different factors for explaining the spatial pattern of ambient heat across European cities. We separately analyse the spatial correlation of each heat metric with four climatological factors (summer mean daily maximum near-surface air temperature $\overline{TX}_{ref}$ and its standard deviation $\sigma_{TX,ref}$ in the reference period 1981-2010; change in summer mean daily maximum near-surface air temperature $\Delta\overline{TX}$ and change in its standard deviation $\Delta\sigma_{TX}$ between 1981-2010 and the model-specific time of 3 °C European warming) and four location factors (latitude, longitude, elevation, flag indicating whether a city is located close to the sea). Summer is defined as the months June, July, and August.

The explanatory variables (i.e., the climatological factors or the location factors) may be correlated, and their contributions cannot be strictly disentangled. We therefore use an approach based on semipartial correlation to quantify the average contribution of each variable to the total explained variance $R^2$ (Schwingshackl et al., 2018). The squared semipartial correlation measures how much of the remaining unexplained variance is explained by an explanatory variable that is introduced after several others have already been considered. If explanatory variables are independent, the sum of the squared semipartial correlation coefficients yields $R^2$. For correlated explanatory variables, the additional contribution of an explanatory variable can be estimated by the average $R^2$ increase of adding the variable to all regression models that contain a subset of the other explanatory variables (Azen and Budescu, 2003; Schwingshackl et al., 2018). If using the averaging method proposed by Azen and Budescu (2003), the sum of all squared semipartial correlations is equal to $R^2$. The variability of the squared semipartial correlation estimates is a measure for collinearities between the explanatory variables and can be used as an uncertainty estimate for the contribution of each explanatory variable. The estimated contribution of each explanatory variable to the spatial variability of each heat metric does not permit statements about causality, as it is purely based on correlation analysis. Instead, the contributions should be interpreted as a measure of the extent to which the explained variables may be influenced by the location of each city or by the climatic conditions and climate change at the location of each city.

### 2.4.2 Relative importance of RCMs and GCMs

We further quantify how much of the variability in ambient heat across the EURO-CORDEX ensemble is due to the choice of GCMs or RCM, respectively. We follow the variance decomposition method of Sunyer et al. (2015) to calculate the variance due to RCMs, due to GCMs, and due to the interaction between RCMs and GCMs. As the interaction term cannot be attributed to either GCMs or RCMs, we interpret it as uncertainty and indicate the contribution of RCMs and GCMs as a range that once includes and once excludes the contribution of the interaction term. For each heat metric, we calculate the percentage contribution of RCMs and GCMs to the total variance across all 72 RCM-GCM model chains.

## 3 Results

### 3.1 Evaluation of the EURO-CORDEX ensemble

To evaluate how well the EURO-CORDEX models reproduce observed temperatures in the 36 major European cities, we compare their temperature distribution to data from E-OBS, ERA5-Land, and weather stations. Figure 3 shows the distributions of summer mean daily maximum near-surface air temperatures in 1981-2010 for all cities as a function of distance from the city centre. Detailed bias distributions for all cities can be found in Supplementary Figure S1, and a map of the multi-model median biases is shown in Supplementary Figure S2. The distribution of the EURO-CORDEX models generally matches the reference data well but is often wider than the distributions of the reference datasets (Figure 3). The EURO-CORDEX simulations reveal a cold bias in many cities lying in the northern and eastern parts of Europe (Dublin, Helsinki, Kazan, Nizhny Novgorod, Oslo, Saint Petersburg, Stockholm), ranging from -1.3 °C to -2.7 °C relative to E-OBS and from -0.3 °C to -1.2 °C relative to ERA5-Land. A warm bias – particularly relative to ERA5-Land – is found for several cities in south-eastern Europe (Belgrade, Bucharest, Kharkiv, Kyiv), ranging from +0.2 °C to +1.0 °C relative to E-OBS and from +1.7 °C to +3.2 °C relative to ERA5-Land. In general, a negative-to-positive tendency from North to South can be identified for the EURO-CORDEX biases (Supplementary Figure S2). ERA5-Land and E-OBS also show systematic differences, with daily maximum temperatures in ERA5-Land being mostly colder than E-OBS and the weather station data. Consequently, the magnitude and sign of the EURO-CORDEX biases strongly depend on the reference dataset. The multi-model median of the EURO-CORDEX ensemble has a warm bias relative to ERA5-Land (+0.5 °C on average across cities) and a cold bias relative to E-OBS (-0.8 °C on average), which is consistent with the findings of Vautard et al. (2021).

The distributions of daily minimum near-surface air temperatures in the EURO-CORDEX models also generally match the reference datasets (Supplementary Figure S3), although the spatial patterns differ from the bias patterns of maximum temperatures (Supplementary Figure S2). Biases are highest in northern, eastern, and southern European cities, while they are lowest in central European cities. The EURO-CORDEX ensemble has a cold bias relative to E-OBS (-0.6 °C on average; most pronounced in Saint Petersburg, Nizhny Novgorod, Copenhagen, Lisbon, Madrid) and to ERA5-Land (-0.8 °C on average; most pronounced in Kazan, Helsinki, Istanbul, Riga, Stockholm). In contrast to the lower daily maximum temperature values in ERA5-Land, daily minimum temperatures in ERA5-Land are warmer than E-OBS in several of the investigated cities.

In some cities, temperatures vary as a function of the distance from the city centre (Figure 3, Supplementary Figure S3). E-OBS shows higher temperatures close to the city centre in Budapest, Prague, and Vienna, while for EURO-CORDEX this is the case in Athens, Brussels, Dublin, Minsk, Munich, Paris, Rome, and Vienna. Yet, these temperature gradients are not necessarily due to UHI but could also be caused by other factors, such as gradients in elevation. For E-OBS and the weather station data, the scarce station density close to the city centres as well as the standard conditions for meteorological measurements (i.e., measurements are taken over grasslands) might be reasons for the lack of pronounced UHI effects. For the other datasets, this might be due to the missing representation of urban areas in the land surface schemes of ERA5-Land and the predominantly rather simple representation of urban areas in the EURO-CORDEX models (Table 1).

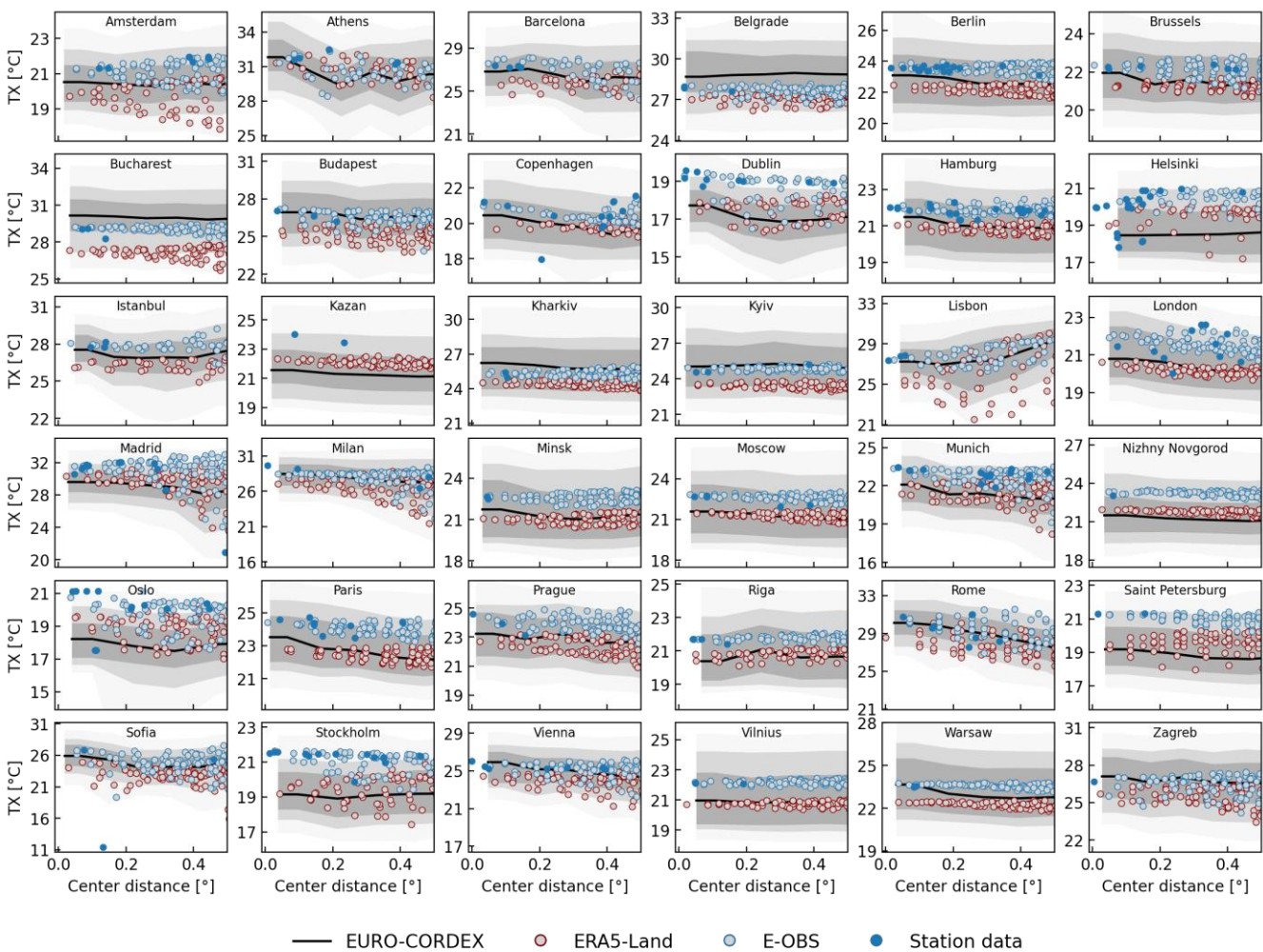

339

**Figure 3:** Distribution of daily maximum near-surface air temperature (TX) in summer for the investigated European cities as function of distance to the city centre. The plot shows summer (June, July, August) average TX over the period 1981-2010 for EURO-CORDEX (black line and grey shading), ERA5-Land (red-edged grey dots), E-OBS (blue-edged grey dots), and station data (filled blue dots). The black line for EURO-CORDEX denotes the multi-model median, dark grey shading the interquartile range across models, and light (very light) grey shading the range from 10th (1st) to 90th (99th) percentile. Only temperatures on land are included (sea areas are masked).

## 3.2 Projections of ambient heat for major European cities

The EURO-CORDEX projections for major European cities show increasing ambient heat under 3 °C European warming with distinct geographical patterns for the three different metrics (Figure 4). Increases in TXx are largest in southern Europe, followed by western and eastern Europe, and lowest in northern Europe. The top five cities in terms of TXx increase (Milan, Madrid, Sofia, Zagreb, Belgrade; numbered from 1 to 5 in Figure 4) are all located in southern Europe but none of them is located close to the sea. Cities in southern Europe located at or close to the sea (e.g., Lisbon, Barcelona, Rome, Athens, Istanbul) also show substantial TXx increase, yet weaker than the cities situated more inland.

The yearly number of days on which TX exceeds 30 °C shows a clear south-to-north gradient, with values being highest in Athens, Madrid, Rome, Bucharest, and Milan (numbered 1 to 5). These cities exceed 30 °C on more than 80 d/y, while the five cities with lowest exceedance rates (all lying in northern Europe; numbered 32 to 36) experience on average less than 2 d/y above 30 °C. Additionally, local climate conditions can play an important role as well, for example in the case of Barcelona, Istanbul, and Sofia, which have lower exceedance rates than the surrounding cities. Varying the threshold level between 25 °C and 33 °C considerably changes the number of yearly exceedance days, but the geographical distribution is not altered much (Supplementary Figure S4).

HWMId-TX is largest in southern European cities, followed by eastern European cities, with values being highest in Barcelona, Madrid, Milan, Sofia, and Rome (numbered 1 to 5). In contrast to the other two metrics, cities located in northern Europe also show high HWMId-TX values (e.g., Oslo, Copenhagen, Stockholm, Helsinki), while lowest HWMId-TX values are projected in an arc spanning from the Netherlands over northern Germany towards the Baltic states.

In several cities, all considered heat metrics show high levels of ambient heat under 3 °C European warming (e.g., Athens, Belgrade, Bucharest, Madrid, Milan, Sofia, Zagreb). For other cities, however, the ambient heat levels differ substantially depending on the metric under consideration. Barcelona, for example, ranks number one in terms of HWMId-TX, but exceeds 30 °C only rarely. Lisbon has substantial increases in TXx and temperatures often exceed 30 °C, but HWMId-TX is rather low. Kazan has substantial increases in TXx and high HWMId-TX values, but TX exceedances above 30 °C are relatively low. Oslo ranks among the cities with weakest changes in TXx and with lowest TX exceedances above 30 °C, but with high HWMId-TX values. These discrepancies may be due to several reasons. For instance, cities with comparatively cooler climate may see large increases in TXx and high HWMId-TX values without having substantial exceedances above 30 °C. Cities with high climatological variability in TXx may have comparatively low HWMId-TX values despite large increases in TXx and, vice versa, relatively low increases in TXx might result in high HWMId-TX values in case of low climatological variability in TXx. Considering only one heat metric might thus lead to unbalanced conclusions about projections of ambient heat for urban areas, potentially underestimating future risks from heat stress.

**TXx change**

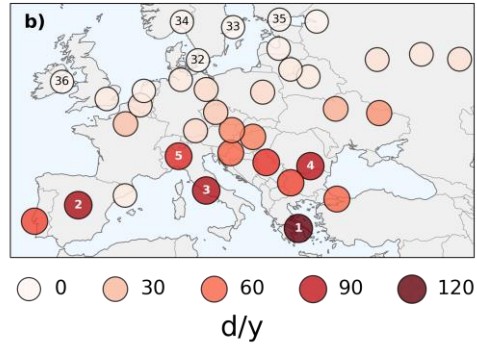

**Exceedance TX>30°C**

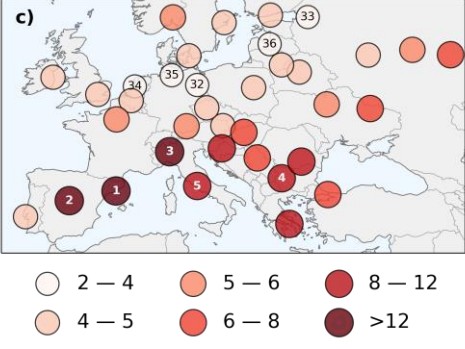

**HWMId-TX**

377

**Figure 4:** Projections of ambient heat at 3 °C European warming according to three different heat metrics for 36 major European cities as simulated by the EURO-CORDEX ensemble. a) Change in yearly maximum near-surface air temperature (TXx) between 1981-2010 and 3 °C European warming, b) TX exceedances above 30 °C at 3 °C European warming, and c) Heat Wave Magnitude Index daily based on TX (HWMId-TX) at 3 °C European warming. The values indicate the multi-model median of the EURO-CORDEX ensemble. Numbers in the circles from 1 to 5 (32 to 36) indicate the five cities with highest (lowest) ambient heat according to each metric.

### 3.3 Identifying factors influencing the spatial patterns of ambient heat across cities

To better understand the spatial patterns of ambient heat projected by the different heat metrics, we estimate how much of the spatial variance is explained 1) by different climate factors, representing each city's temperature climatology as well as its projected changes, and 2) by different location factors (Figure 5; see Section 2.4.1 for methodological details). Generally, the considered climate factors ($\overline{TX}_{ref}$, $\sigma_{TX,ref}$, $\Delta\overline{TX}$, and $\Delta\sigma_{TX}$; see Section 2.4.1 for their definition) explain more of the spatial patterns than the location factors (latitude, longitude, elevation, location close to sea). Regarding climate factors (Figure 5a), the spatial pattern of TXx change is mostly influenced by the climate factors $\Delta\overline{TX}$ and $\Delta\sigma_{TX}$, while climate conditions in the reference period do not contribute significantly. For TX exceedances above 30 °C, the maximum temperature in the reference period contributes by far the most, followed by $\Delta\overline{TX}$. For HWMId-TX, the strongest contributions stem from $\Delta\overline{TX}$ and $\sigma_{TX,ref}$. Regarding location factors (Figure 5b), latitude, longitude, and whether a city is located close to the sea partly explain the spatial pattern of TXx change, albeit with rather low model agreement. For the TX exceedances above 30 °C, latitude plays the dominant role, while the contributions of all other factors remain negligible. For HWMId-TX, the explanatory power of all location factors remains low, with latitude being the only factor that explains some of the signal.

Across the three metrics, most of the spatial variability can be explained for the TX exceedances above 30 °C ($R^2$=0.78 for climate and $R^2$=0.59 for location factors; considering only variables with significant contribution in at least 50% of the EURO-CORDEX models), followed by TXx change ($R^2$=0.58 for climate and $R^2$=0.50 for location factors), while the explained variance of the spatial patterns of HMWId remains rather low ($R^2$=0.42 for climate and $R^2$=0.19 for location factors). The contribution of the single climate factors depends strongly on the selected metric, whereas for location factors only latitude plays a major role. All other location factors – despite being statistically significant in some cases – only contribute little to the total variance explained. The high uncertainty for the contribution of some explanatory variables (e.g., $\Delta\overline{TX}$ and $\Delta\sigma_{TX}$ for TXx change, $\overline{TX}_{ref}$ and $\Delta\overline{TX}$ for TX exceedances above 30 °C) points to collinearities between these explanatory variables, which can, however, not be disentangled based on correlation analysis.

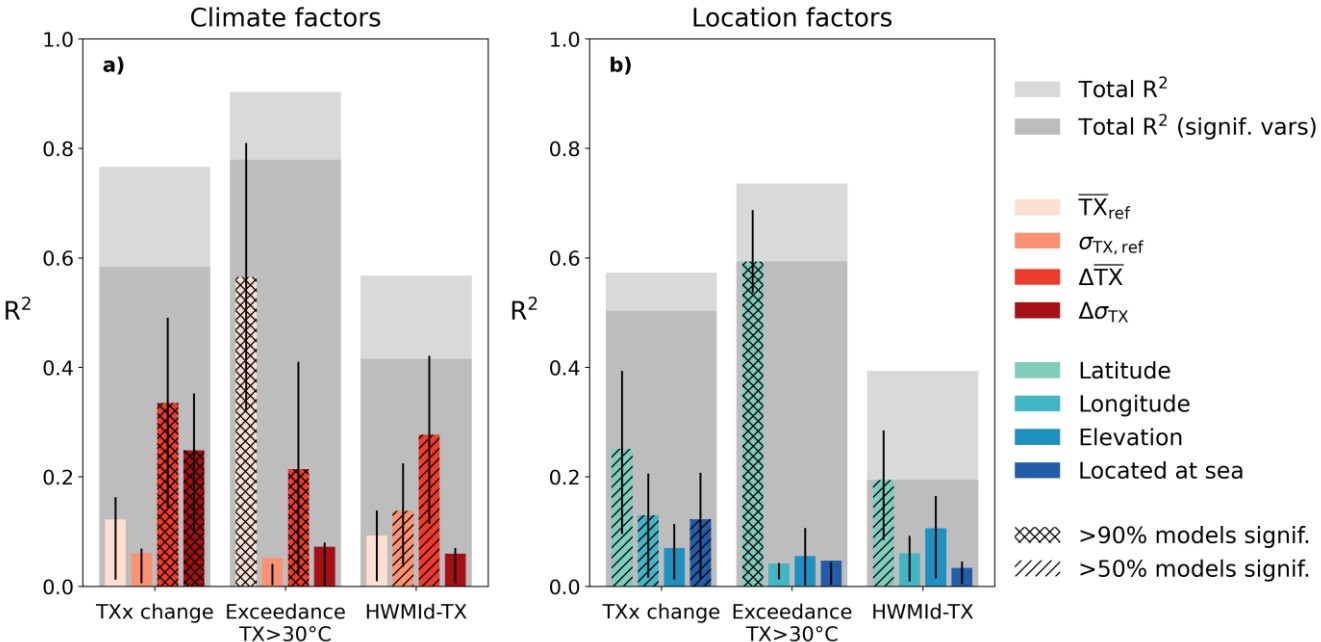

**Figure 5:** Contribution of different explanatory variables to the explained variance ($R^2$) of the spatial patterns of ambient heat across European cities in the EURO-CORDEX ensemble. Explanatory variables are divided into a) climate factors (summer mean daily maximum near-surface air temperature $\overline{TX}_{ref}$ and its standard deviation $\sigma_{TX,ref}$ in the reference period; change in summer mean daily maximum near-surface air temperature $\Delta\overline{TX}$ and its standard deviation $\Delta\sigma_{TX}$ between the reference period 1981-2010 and 3 °C European warming) and b) location factors. Coloured bars denote the median estimate for each factor, black whiskers denote the uncertainty indicated as interquartile range (calculated from the pooled data of all 72 EURO-CORDEX models and eight regression models). Hatching with lines (crosses) indicates whether at least 50% (90%) of the EURO-CODEX models indicate statistically significant contribution of the respective explanatory variable (Student's t-test, $p<0.05$). Background bars coloured in light grey indicate total $R^2$ considering all explanatory variables, background bars in dark grey indicate total $R^2$ if considering only explanatory variables that are statistically significant in at least 50% of the EURO-CORDEX models (Student's t-test, $p<0.05$). The contribution of each climate/location factor is estimated by semipartial correlation (see Section 2.4.1).

## 3.4 Comparing projections of ambient heat during daytime and nighttime

The results presented so far are based on daily maximum temperature and are thus mostly indicative for daytime conditions. We additionally consider daily minimum temperature (TN) to investigate projections of ambient heat during nighttime, which play an important role for human health as well, since elevated nighttime temperatures can reduce people's capacity to recover and thus weaken their physical conditions (Royé et al., 2021; Thompson et al., 2022). The geographical patterns of the TN-based heat metrics are generally similar to the TX-based patterns (Figure 6) with highest levels of ambient heat in southern European cities. Yet, several distinct differences are evident. The TNx increase is generally smaller than the TXx increase, except for cities located at the Baltic Sea, which exhibit a stronger increase in TNx than TXx. Days with TN>20 °C ("tropical nights") are rarer than days with TX>30 °C, except for Barcelona and Istanbul, both of which having substantially more days with TN>20 °C than TX>30 °C (note that no bias adjustment was applied neither for TN>20 °C nor for TX>30 °C; bias-adjusting the mean of the TN distribution based on ERA5-land data even increases the days with TN>20 °C in Barcelona and Istanbul; not shown). In northern Europe, days with TN>20 °C or TX>30 °C both occur very rarely, and differences are thus negligible. Varying the TN threshold level between 15 °C and 23 °C considerably changes the number of yearly exceedance days, but the geographical distribution is not altered much (not shown). HWMId-TN shows much higher values than HWMId-TX, particularly in southern European cities but also in central European cities and in several cities located at the Baltic Sea. Differences between HWMId-TN and HWMId-TX are particularly large in Istanbul, Barcelona, and Rome. The higher HWMId-TN values suggest that nighttime heatwaves will become more severe than daytime heatwaves in the investigated cities as compared to the typical nighttime and daytime climate conditions of the recent past (1981-2010).

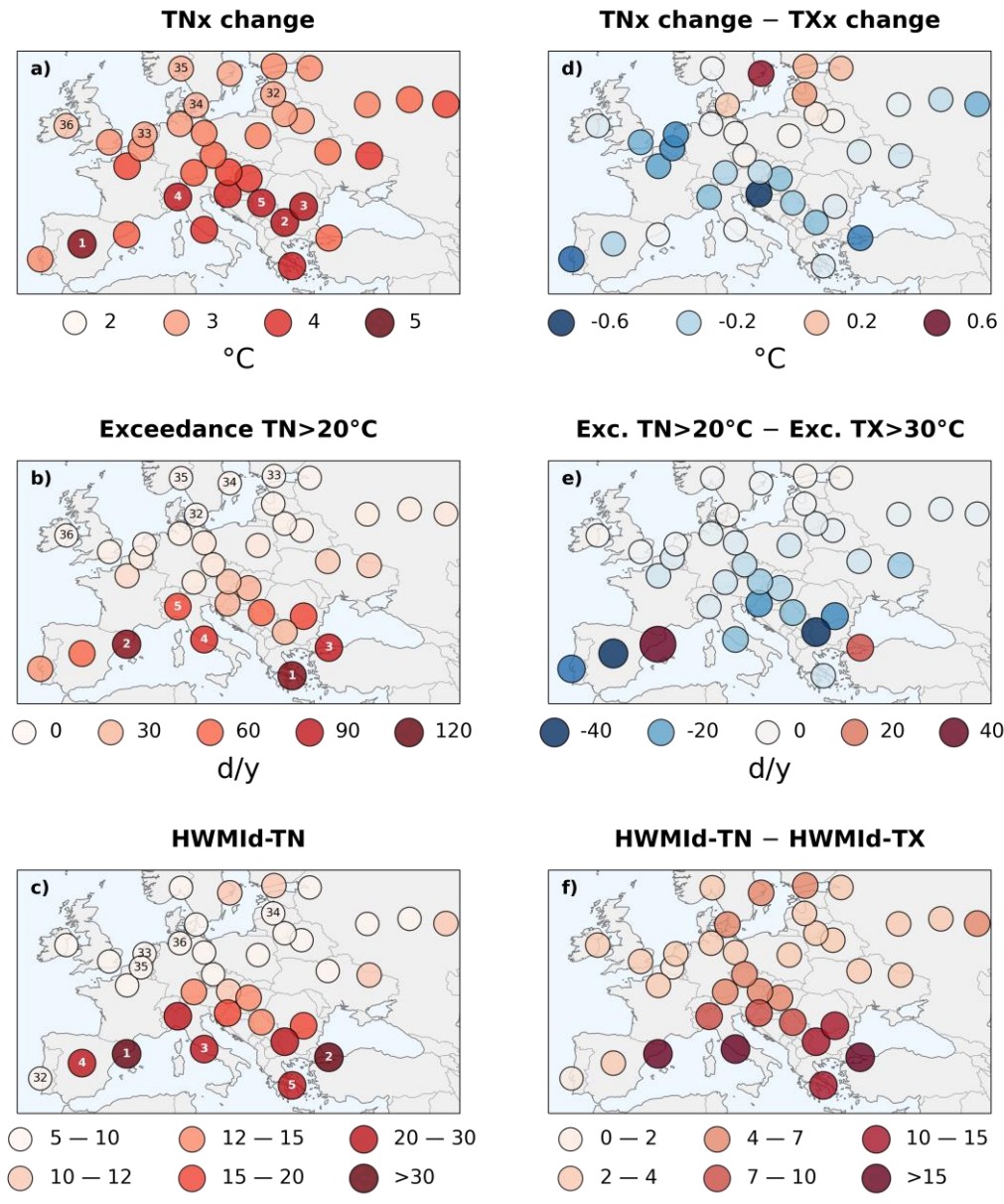

439

**Figure 6:** As in Figure 4 but for daily minimum near-surface air temperature (TN) in panels (a) - (c). Panels (d) - (f) show the difference between ambient heat estimates based on TN and based on daily maximum near-surface air temperature (TX). Note that the scale for HWMId-TN differs from the HWMId-TX scale in Figure 4.

**3.5 EURO-CORDEX projections of ambient heat in comparison to CMIP5 and CMIP6 projections**

We further compare the projections of ambient heat by the EURO-CORDEX, CMIP5, and CMIP6 ensembles for the 36 European cities (Figure 7). The general patterns of CMIP5 and CMIP6 reflect the results of Figure 4, showing a strong TXx increase in south-eastern and eastern European cities, high TX exceedance rates of 30 °C in southern and some eastern European cities, and high HWMId-TX values in southern and some northern European cities (note the logarithmic axis for the latter). In terms of TXx change, the CMIP5 and CMIP6 ensembles generally project a stronger increase in ambient heat than the EURO-CORDEX models, particularly in south-eastern, eastern, and north-eastern European cities, while, for Lisbon, Athens, and Istanbul, the EURO-CORDEX ensemble projects stronger TXx increases. Regarding TX exceedances above 30 °C, the EURO-CORDEX ensemble projects much higher exceedance rates than the CMIP5 and CMIP6 ensembles in southern European cities (e.g., Lisbon, Milan, Athens, Istanbul), whereas the CMIP5 and CMIP6 ensembles show larger exceedance rates in north-eastern European cities and in Barcelona. The CMIP5 and CMIP6 ensembles project higher HWMId-TX values in almost all cities except Madrid, Nizhny Novgorod, and Kazan. Differences in HWMId-TX between the CMIP5 and CMIP6 and EURO-CORDEX ensembles are particularly pronounced in Stockholm, Rome, Athens, and Istanbul. The projected geographical patterns of ambient heat from the CMIP5 and CMIP6 ensembles are generally similar; notable differences are only found for TX exceedances above 30 °C, where CMIP6 has substantially higher values in southern European cities whereas CMIP5 shows more exceedances in northern European cities.

To investigate the effect of dynamical downscaling by RCMs, we additionally consider the projections of ambient heat by the EURO-CORDEX GCM ensemble (dashed purple line in Figure 7; see Section 2.1.2 for its definition). The EURO-CORDEX GCM ensemble resembles more closely the results of the CMIP5 ensemble than of the EURO-CORDEX ensemble, except for some cities (e.g., Amsterdam, Copenhagen, Stockholm, Saint Petersburg, Nizhny Novgorod for TXx changes; Rome for TX exceedances above 30 °C; Lisbon for HWMId-TX). In combination with the fact that the EURO-CORDEX GCM ensemble shows very similar ESAT trends to the EURO-CORDEX RCM ensemble (Figure 2a), this indicates that differences in projections of ambient heat between the EURO-CORDEX and CMIP5 ensembles are mostly connected to the dynamical downscaling by RCMs. For cities located close to mountains (e.g., Athens) or close to the sea (e.g., Lisbon, Barcelona, Stockholm), the higher spatial resolution of RCMs should thus deliver more accurate estimates than the more coarsely resolved GCMs. This is reflected in the large differences between CMIP5 and EURO-CORDEX estimates for several cities, particularly for TX exceedances above 30 °C and for HWMId-TX.

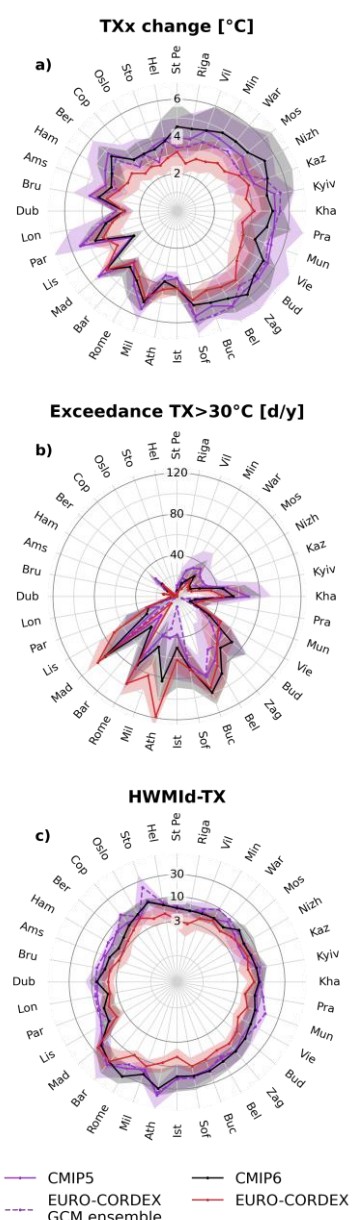

**Figure 7:** Projections of ambient heat in European cities for EURO-CORDEX, CMIP5, CMIP6, and the EURO-CORDEX GCM ensemble. Cities are arranged according to their geographical location, i.e., northern European cities at the top, eastern European cities on the right, southern European cities at the bottom, and western European cities on the left. a) Change in yearly maximum near-surface air temperature (TXx) between 1981-2010 and 3 °C European warming, b) TX exceedances above 30 °C at 3 °C European warming, c) Heat Wave Magnitude Index daily based on TX (HWMId-TX) at 3 °C European warming. Note the logarithmic axis for the HWMId-TX panel. Lines indicate the multi-model median and shading the interquartile range across models.

### 3.6 Uncertainty of ambient heat projections

To evaluate the robustness of our results, we estimate how strongly the estimates of ambient heat vary across the EURO-CORDEX models and how much they change in space, that is, within a box of 3x3 grid cells around the grid box located closest to the city centres. The large ensemble of 72 GCM-RCM combinations enables a thorough assessment of the model uncertainty, which we quantify here as the interquartile range (IQR) across models (Figure 7). Uncertainties of TXx change lie between 1 °C and 2 °C in almost all cities, with uncertainties being lowest in southern European cities (where uncertainties are ~1 °C). For TX exceedances above 30 °C, we calculate relative uncertainties (IQR divided by multi-model median; not shown) to reflect the large variability of exceedance rates across cities. The relative uncertainties of TX exceedances above 30 °C are lowest in southern European cities (between 20% and 60%) except for Barcelona, where the relative uncertainty is larger than 300% (and the distribution is skewed towards higher values). In contrast to the other metrics, the uncertainties of HWMId-TX are higher in southern European cities (uncertainties lying between 4 and 8) than in northern European cities (uncertainties lying between 2 and 6), with uncertainties being highest in Barcelona (IQR = 32) followed by Madrid (IQR = 13).

To quantify the spatial variability of ambient heat, we calculate the heat metrics individually for each grid cell in a box of 3x3 grid cells around the city centres. The spatial variability is quantified by how much ambient heat varies within the 3x3 grid cells (Supplementary Figure S5). In the large majority of cities, the TXx change estimates remain very similar if using the 3x3 box, indicating that the estimated trends in TXx do not change much within the grid cells surrounding the city centres. Lisbon, Barcelona, Athens, Helsinki, and Istanbul are the cities with the largest spatial variability in TXx changes. Regarding TX exceedances above 30 °C, the largest variabilities are found in Lisbon, Barcelona, Athens, Istanbul, Rome, and Sofia. HWMId-TX values show very large spatial variability in Barcelona and Helsinki, and pronounced variability in Istanbul, Copenhagen, Athens, and Dublin. If only considering grid cells with land fractions larger than 25%, 50%, or 75%, the variability decreases substantially in almost all cities with large spatial variability in heat metrics. This suggests that ambient heat strongly differs between land and sea areas, particularly for HWMId-TX and for TX exceedances above 30 °C. For HWMId-TX this might be due to the higher TXx variability over land areas than over the sea in the reference period 1981-2010 (Supplementary Figure S6), resulting in much larger HWMId-TX values over sea than over land. Consequently, cities located close to the sea might be affected by this stark land-sea contrast, particularly if their climate is strongly influenced by the sea.

We further test how TX exceedances above 30 °C in the grid cell closest to the centre of each city change if applying a simple adjustment method that 1) adjusts the mean of each EURO-CORDEX model to the mean of the ERA5-Land data and 2) adjusts both the mean and the standard deviation (Supplementary Figure S7, see also Section 2.3 for methodological details). The most striking effect of adjusting the data is a reduced uncertainty of the projected TX exceedances above 30 °C. Moreover, the adjusted exceedance rates are statistically significantly lower in 13 cities and higher in 2 cities if only the mean is adjusted (Kolmogorov-Smirnow test, p<0.05); and lower in 15 cities and higher in 6 cities if both mean and standard deviation are adjusted. In the remaining cities, the differences are not statistically significant. The effects of the simple adjustment method

are largest in Lisbon, Rome, Sofia, and Bucharest with substantially lower exceedance rates in case of adjustment. Adjusting
only the mean or adjusting both mean and standard deviation generally yields similar results (differences are largest in Istanbul
and Lisbon) with the latter method tending to yield lower exceedance rates.
The rather complete matrix of RCM-GCM combinations enables us to quantify how much of the variability in ambient heat
across the EURO-CORDEX models is due to the choice of GCMs or RCMs (Figure 8, see section 2.4.2 for methodological
details). The variability across all RCM-GCM combinations is mostly due to RCMs (60% to 75% for TXx change, 60% to
70% for TX exceedances above 30 °C, and 50% to 65% for HWMId-TX), highlighting that the downscaling by RCMs plays
a crucial role for the ambient heat estimates in urban areas. Additionally, several patterns can be identified for certain RCMs
and GCMs, which indicates that the choice of RCMs and GCMs is also important. Among RCMs, projections of ambient heat
in terms of TXx change and HWMId-TX are highest for HadREM3-GA7-05, and in terms of TX exceedances above 30 °C
values are highest for WRF381P, HadREM3-GA7-05, and ALADIN63. Comparatively low increases in ambient heat are
projected by the RCMs HIRHAM5, RACMO22E, and COSMO-crCLIM-v1-1. Differences between GCMs are less
pronounced. Projections of ambient heat are highest for NorESM1-M and CanESM2 in terms of TXx change, for CanESM2,
HadGEM2-ES, and MIROC5 in terms of TX exceedances above 30 °C, and for NorESM1-M, CanESM2, and MIROC5 in
terms of HWMId-TX. It should be noted though that the results for CanESM2 and MIROC5 might be less robust as each of
them is only used twice as driving GCM. Comparatively low increases in ambient heat are projected by CNRM-CM5 and
IPSL-CM5A-MR for TXx change, by EC-EARTH and CNRM-CM5 for TX exceedances above 30 °C, and by CNRM-CM5
and MPI-ESM-LR for HWMId-TX.


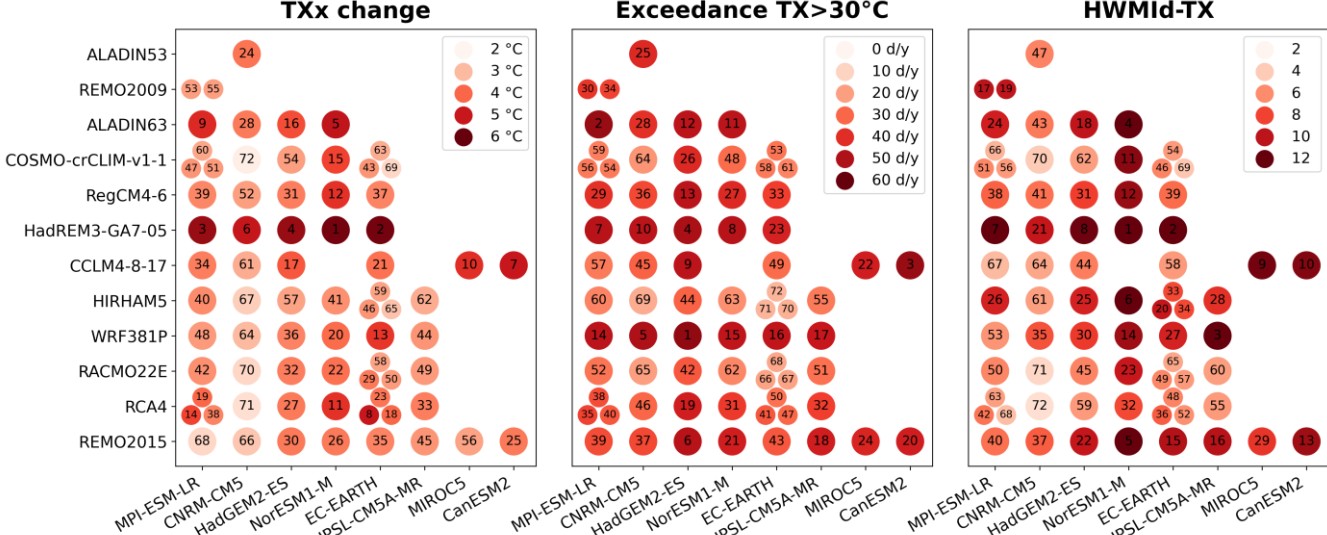


**Figure 8:** GCM-RCM matrix of EURO-CORDEX models for the change in yearly maximum near-surface air temperature (TXx) between 1981-2010 and 3 °C European warming, b) TX exceedances above 30 °C at 3 °C European warming, and c) Heat Wave Magnitude Index daily based on TX (HWMId-TX) at 3 °C European warming. Each circle indicates the average value across all investigated cities for each individual EURO-CORDEX model. Numbers in the circle indicate the ranking of models from 1 (highest ambient heat) to 72 (lowest ambient heat). Multiple ensemble members for a GCM-RCM combination are indicated as smaller circles.

## 4 Discussion

### 4.1 Interpretation and implications of results

All three analysed heat metrics show strong increases in ambient heat in southern European cities at 3 °C European warming. Substantial increases in ambient heat are also evident in other European regions; yet, the spatial patterns strongly depend on the metric under consideration. TXx increases considerably in western and eastern Europe, TX exceedances above 30 °C show a clear south-to-north gradient with almost no exceedances in northern European cities, and HWMId-TX yields comparatively high values in eastern and northern European cities. This has implications for the estimation of future heat stress, as the projected outcomes can vary strongly depending on the considered metric. For instance, regions in northern Europe that are usually not considered as very prone to heat stress show relatively high values of HWMId-TX. Since health impacts do not only depend on universal physiological limits but also on the climate conditions people are used to (Petkova et al., 2014; Åström et al., 2013), metrics considering the climatology of a region (such as HWMId-TX) can give important insights into the risk of future heat stress that might otherwise be missed. This also concerns nighttime conditions, as HWMId-TN is even higher than HWMId-TX (Figure 6).

The identified spatial patterns broadly agree with results of other studies, showing an increase in heatwave risk in southern Europe along with substantial increases in coastal regions in northern Europe (Guerreiro et al., 2018; Smid et al., 2019; Lin et al., 2022) – as we find for HWMId-TX – and a clear south-to-north gradient in exceedances of WBGT>28 °C (Casanueva et al., 2020) – consistent with the patterns of TX exceedances above 30 °C. Guerreiro et al. (2018) found that temperatures during heatwaves increase strongest in central Europe, while the TXx increases estimated in our study are highest in southern European cities. This discrepancy between the findings of Guerreiro et al. (2018) and our results could, on the one hand, be related to the fact that TXx does not directly reflect temperatures during heatwaves. On the other hand, it could also be due to the more pronounced increase of extreme temperatures in central Europe in CMIP5 compared to EURO-CORDEX (Supplementary Figure S8). Supplementary Figure S8 also shows that the EURO-CORDEX models project an amplified warming of the Baltic Sea compared to the surrounding land areas, which is likely the reason for the high values of HWMId-TX in northern European coastal cities.

In many of the investigated cities, CMIP5 and CMIP6 project higher increases in TXx and larger HWMId-TX values than EURO-CORDEX. This is likely caused by discrepancies in external forcing data and differences in process implementation (see Section 2.1.2). Specifically, the CMIP5 and CMIP6 simulations are based on future scenarios with decreasing atmospheric aerosol concentrations over the European domain, while the EURO-CORDEX simulations assume a constant atmospheric aerosol load (Boé et al., 2020). The RCMs of EURO-CORDEX may thus underestimate future warming in Europe as they do not consider the amplified warming from the additional solar radiation reaching and heating the Earth's surface in Europe because of the decreasing aerosol concentrations. In addition, unlike CMIP5 and CMIP6 GCMs, several RCMs do not consider plant physiological effects (Schwingshackl et al., 2019). The closing of plant stomata due to higher $CO_2$ concentrations and the associated decrease in latent and increase in sensible heat fluxes, which lead to enhanced extreme temperatures, are thus

not fully captured by RCMs. These differences between GCMs and RCMs suggest that RCMs likely underestimate future levels of ambient heat in European cities. Yet, for several southern European cities the EURO-CORDEX models project considerably more days exceeding 30 °C than CMIP5 and CMIP6. In coastal cities, such as Istanbul, Athens, and Lisbon, these differences are likely due to the higher spatial resolution of EURO-CORDEX, which enables a better distinction of land and ocean grid cells. In other cities, like Madrid or Rome, better resolved orography might be the reason for the more frequent exceedances in EURO-CORDEX. Yet the causes for some discrepancies remain unclear, for instance for the more frequent exceedances above 30 °C projected by EURO-CORDEX for Milan, which lies in the rather flat Po Valley, or for the coastal city Barcelona, where EURO-CORDEX shows much fewer exceedances above 30 °C than CMIP5 and CMIP6.

In some cities, the ranking varies considerably depending on the considered heat metric (particularly in Barcelona, Oslo, Lisbon, Warsaw, and Berlin; Figure 4), indicating that the choice of metrics may strongly influence projections of ambient heat in these cities. These discrepancies in the ambient heat estimates from different heat metrics depend, for instance, on the local climate conditions, as the number of days exceeding 30 °C is strongly connected to the average summer temperatures in a city (see Figure 5a) and HWMId values are influenced by the local temperature variability (see Eq. (1)). Additionally, in some cities the projections vary considerably within a box of 3x3 grid cells around the city centre (Supplementary Figure S5), especially for TX exceedances above 30 °C and HWMId-TX. The variability is generally largest for cities located close to the sea, particularly for HWMId-TX. This is related to the fact that HWMId-TX values are generally much higher over the sea than on land, which is mostly due to the low climatological variability of TXx over the sea (Supplementary Figure S6). If cities are located close to the sea, the estimated HWMId-TX values may thus strongly depend on how much of the grid cell located closest to the city centre is covered by land and on how much this land fraction varies across EURO-CORDEX models. In such cases, an accurate representation of local interactions between land and sea (e.g., higher spatial resolution, accurate representation of advection, consideration of humidity) is necessary to generate more robust projections of ambient heat.

The spatial patterns of the heat metrics can largely be explained by the local temperature climatology and its projected changes (see importance of climate factors in Figure 5), with varying importance of the single explanatory factors depending on the considered metric. The explanatory factors explain most of the spatial variability in TXx change and in TX exceedances above 30 °C but they only partly explain the spatial variability in HWMId-TX. The remaining unexplained variance of the heat metrics might be connected to the amplified increase of extreme temperatures (Seneviratne et al., 2016; Vogel et al., 2017) (we use summer mean TX as explanatory factor) or asymmetric changes in the temperature distributions (we use the symmetric standard deviation of TX as explanatory factor). For HWMId-TX, the relatively large unexplained variance might be specifically connected to the definition of HWMId, i.e., to the usage of a cut-off temperature to define heatwaves and to the standardisation based on the climatology of TXx. The same is the case for TX exceedances above 30 °C, which are generally non-linear due to the usage of the absolute threshold of 30 °C. Among the location factors, the latitude of a city is the most important factor for explaining the spatial variance, particularly for TX exceedances above 30 °C. Generally, the explained variance is lower for location factors than for climate factors, indicating that local climate does certainly not only depend on the coordinates and elevation of a location but also on other local factors, such as the predominant atmospheric circulation or

local feedbacks (e.g., vegetation, soil moisture). As the contribution of the explanatory variables to the explained variance is
quantified based on correlation analysis, definitive cause-effect chains cannot be deduced. Particularly for the climate factors,
the results should thus rather be interpreted as an indication of the extent to which the calculated heat metrics are influenced
by the underlying temperature distribution and its projected future change.

## 4.2 Limitations and potential improvements

The ~12.5 km spatial resolution of the EUR-11 simulations enables a much more detailed assessment of climate variability
and climate change at the city-level compared to GCMs, which have a much coarser spatial resolution (~100 km). Yet, most
land surface modules of models in the 0.11° EURO-CORDEX ensemble only employ a simplified representation of urban
areas (Table 1), which prevents the full exploitation of their high spatial resolution for studies focusing on urban areas. A few
models represent urban areas as rock surfaces, thus neglecting the influence of urban vegetation on the surface energy balance
and the influence of urban buildings on turbulence, radiation, and hydrology. Other models apply adjusted parameters (e.g.,
for albedo and roughness length) and a reduced vegetation cover in urban areas, and thus consider the characteristics of cities
to some extent. One of the models uses a sophisticated urban land model, which includes various aspects of urban areas, such
as urban canyons, different levels of urbanisation, and radiation and hydrology schemes specifically adapted for urban areas.
Despite these substantial differences in how urban areas are represented, no direct link can be found between the general
behaviour of the different models in the projection of ambient heat (e.g., comparatively high levels of ambient heat in
HadREM3-GA7-05 and WRF381P, and comparatively low levels in HIRHAM5, RACMO22E, and COSMO-crCLIM-v1-1,
with all of these models using the adjusted-parameter approach to represent urban areas) and their representation of urban
areas (Figure 8, Table 1). The CORDEX Flagship Pilot Study on URBan environments and Regional Climate Change (URB-
RCC) is tackling the question of urban parameterizations and may provide important advancements for urban-resolving climate
modelling in the medium term. Investing in the development of urban parameterisations might have further benefits, as their
implementation in climate models may also affect regional climate outside the urban areas (Katzfey et al., 2020). Furthermore,
urban temperatures usually exhibit large variability within a city, i.e., at scales that currently cannot be resolved by the 0.11°
EURO-CORDEX ensemble. Urban-resolving climate modelling may provide a way forward to better quantify climate effects
at scales resolving single neighbourhoods (Sharma et al., 2021; Hamdi et al., 2020), which would add valuable information
for assessing the risk of heat stress due to climate change at scales relevant for local health authorities and city planners.
The reanalysis ERA5-Land does not have a dedicated urban tile either, which reduces its suitability for analysing climate at
city-level despite its high resolution of about 9 km. Moreover, the missing urban representation currently prevents the usage
of ERA5-Land as a reference dataset for the application of bias adjustment to investigate urban climate. Climate data from E-
OBS might reflect urban conditions to the extent weather stations are present in cities. However, weather stations are located
on grassland, and E-OBS might thus underestimate ambient heat in heavily sealed parts of cities, such as city centres, inner-
city residential areas, or industrial zones. In case data from paired weather stations inside a city and in its rural surroundings
are available, a bias adjustment procedure for urban areas developed by Burgstall et al. (2021) can be applied to adjust climate
model data to urban conditions.
In our analysis, we do not find pronounced UHI effects (Figure 3, Supplementary Figure S3), which is likely related to the
simplified representation of urban areas in RCMs. UHI may additionally increase in the future due to global warming (Koomen
and Diogo, 2017; Tewari et al., 2019) and urban expansion (Huang et al., 2019; Koomen and Diogo, 2017), and UHI can
further be elevated during heatwaves (Ward et al., 2016). More sophisticated representations of urban areas in RCMs would
make it possible to assess how the EURO-CORDEX models project future UHI developments, and could facilitate sensitivity
studies to identify the contributions of climate change, local climate feedbacks, and urbanisation to the projected increase of
ambient heat in cities.
Differences in climate forcing or process implementation between the CMIP5, CMIP6, and EURO-CORDEX ensembles, such
as differences in aerosol forcing (Boé et al., 2020; Gutiérrez et al., 2020; Nabat et al., 2020) or diverging trends in cloudiness
(Bartók et al., 2017), might further explain discrepancies in climate projections (Taranu et al., 2022). Additionally, several
EURO-CORDEX models do not consider plant physiological $CO_2$ effects and thus likely underestimate extreme temperatures
(Schwingshackl et al., 2019). Although the latter effect is confined to vegetated surfaces and should thus be less relevant in
heavily sealed urban areas, it might still influence urban temperatures in RCMs that consider vegetation in their representation
of urban areas. This might partly explain the lower ambient heat projections of the EURO-CORDEX ensemble compared to
the CMIP5 and CMIP6 ensembles, particularly in eastern and northern Europe.
The usage of absolute thresholds for estimating the number of exceedance days (i.e., 30 °C for daily maximum temperature
and 20 °C for daily minimum temperature) does not reflect that temperatures vary considerably across European cities.
Consequently, the number of exceedance days differs substantially across cities, showing a strong gradient from southern to
northern European cities. While absolute temperature thresholds are a common metric used for projections of ambient heat
(e.g., Schwingshackl et al., 2021; Zhao et al., 2015; Kjellstrom et al., 2009; Casanueva et al., 2020), epidemiological studies
show continuous increases in health impacts above the locally optimal temperature (i.e., the temperature where minimal effects
of health outcomes are observed, Gasparrini et al., 2015). Moreover, epidemiological studies increasingly use temperature
percentiles as exposure metric instead of absolute temperatures to better reflect local conditions (Masselot et al., 2023).
**5 Conclusions**
EURO-CORDEX simulations at 0.11° resolution (EUR-11, ~12.5 km) deliver climate data for Europe at a resolution that is
high enough to analyse projections of ambient heat at the city-level (Figure 1). The temperature distributions of the EURO-
CORDEX models generally agree with data from ERA5-Land and E-OBS in the 36 major European cities investigated, despite
a slight TX warm bias compared to ERA5-Land, a slight TX cold bias compared to E-OBS, and a TN cold bias relative to both
ERA5-Land and E-OBS (Figure 3, Supplementary Figure S3).

Using three different metrics to quantify ambient heat at 3 °C warming in Europe relative to 1981-2010 (i.e., changes in TXx, number of days with temperatures exceeding 30 °C, and HWMId), we find that ambient heat is projected to increase throughout the 36 major European cities investigated. Southern European cities will be particularly affected by high levels of ambient heat, but depending on the considered metric, cities in central, eastern, and northern Europe may also experience substantial increases in ambient heat (Figure 4). Nighttime HWMId increases even more strongly than daytime HWMId (Figure 6), with potentially severe implications for health (He et al., 2022). In several cities, the projected levels of ambient heat strongly depend on the considered metric, such as in Barcelona, Oslo, Lisbon, and Warsaw. This indicates that estimates based on a single metric might not appropriately reflect the risks of adverse health effects due to ambient heat in a warmer climate.

We further analyse the spatial patterns of the ambient heat projections in light of the underlying temperature climatology and its projected changes and the location of the different cities (Figure 5). Changes in TXx are mostly connected to projected changes in the mean and variability of TX, TX exceedances above 30 °C depend mostly on the average TX value in the reference period and its projected change, and the spatial patterns of HWMId are partly explained by changes in TX and the variability in the reference period. Regarding the location of cities, latitude plays the predominant role for explaining the spatial patterns, while the other factors (longitude, elevation, location close to sea) only have limited explanatory power.

The EURO-CORDEX ensemble estimates lower increases in TXx and lower HWMId values than the CMIP5 and CMIP6 ensembles in the majority of the analysed cities (Figure 7). Yet, the EURO-CORDEX ensemble has higher TX exceedance rates of 30 °C in several cities, particularly in southern Europe. This discrepancy can be due to several factors, such as differences in forcing (Boé et al., 2020; Gutiérrez et al., 2020; Nabat et al., 2020), differences in process implementation (e.g., Bartók et al., 2017; Schwingshackl et al., 2019; Taranu et al., 2022), or the higher spatial resolution of EURO-CORDEX models being able to better represent local climate conditions. Yet, several EURO-CORDEX models employ a rather simple representation of urban areas (Table 1), and the specific climate conditions in urban areas are thus not fully captured.

The large ensemble of 72 EURO-CORDEX simulations enables a thorough uncertainty assessment, quantified by the spread across models. The uncertainties of TXx change are generally relatively low (around 1 °C to 2 °C in all cities). For TX exceedances above 30 °C, relative uncertainties range from 20% to 60% in most southern European cities but are higher in northern European cities due to their lower TX exceedance rates of 30 °C. Applying a simple adjustment (see Section 2.3) reduces the uncertainties of the projected TX exceedances above 30 °C in all cities and yields lower exceedance rates in about 40% of the cities. The estimates of ambient heat show high spatial variability around the city centre in cities located close to the shore. Particularly for HWMId, the estimates differ substantially depending on the presence of water or land in the respective grid cell (Supplementary Figure S5). Accurate representations of land and sea and of their interplay are thus essential for quantifying ambient heat in coastal cities.

Our analysis provides an important contribution to estimate ambient heat in 36 major European cities by considering three different metrics and using data from high-resolution RCM simulations. Future studies would benefit from a more comprehensive representation of urban areas in models, which might be developed by the CORDEX Flagship Pilot Study on URBan environments and Regional Climate Change (URB-RCC) for RCMs. Improving the representation of urban areas in

the land surface modules of the EURO-CORDEX RCMs and including an urban representation in ERA5-Land would allow
for an even more accurate estimation of ambient heat at the city-level. Further, the coupling of urban canopy layer models with
regional climate models might pave the way for detailed analyses of heat stress in cities by combining the high spatial
resolution of urban canopy layer models with the climate variability estimates from RCMs. Such analyses could provide an
important step forward towards a comprehensive analysis of ambient heat in European cities and worldwide, and it could be
combined with estimates of exposure and vulnerability to comprehensively quantify future risk of heat extremes.
Cities are expected to increasingly become climate hotspots due to their high population density and the local climate
conditions that are partly influenced by how cities are structured. At the same time, their large innovation potential also gives
cities the opportunity to lead the way in implementing climate adaptation strategies. Providing detailed and accurate data about
ambient heat projections at the city-level is essential to enable cities to plan specific and effective adaptation measures against
future heat extremes.

**Code availability**
The programming code used for the analyses and for creating the figures is available on https://github.com/schwings-
clemens/ambient-heat-european-cities.
**Data availability**
Data supporting this study is publicly available from https://doi.org/10.5281/zenodo.8043755. EURO-CORDEX, CMIP5, and
CMIP6 data is available via the Earth System Grid Federation (ESGF) and can be downloaded from https://esgf-data.dkrz.de.
ERA5-Land is available from https://doi.org/10.24381/cds.e2161bac. E-OBS is available from
https://doi.org/10.24381/cds.151d3ec6. Weather station data from GSOD can be retrieved from https://data.nodc.noaa.gov/cgi-
bin/iso?id=gov.noaa.ncdc:C00516 and weather station data from ECA&D can be retrieved from https://www.ecad.eu.
**Author contributions**
CS and JS conceptualised the study. CI and CS curated the data. CS developed the methodology, performed the analysis, and
created the visualisations. JS and KA acquired funding. CS and AD drafted the manuscript. CS, AD, CI, KA, and JS edited,
wrote, and revised the manuscript.

**Competing interests**

The authors declare that they have no conflict of interest.

**Acknowledgements**

We thank Nina Schuhen for her support with the statistical analysis, particularly regarding the quantification of how much the different explanatory factors can explain the observed spatial patterns of ambient heat. We further thank Marit Sandstad for processing the ERA5-Land data. We acknowledge the E-OBS dataset and the data providers in the ECA&D project (https://www.ecad.eu). This study contains modified Copernicus Climate Change Service information [2022]. We acknowledge the World Climate Research Programme's Working Group on Coupled Modelling, which is responsible for CMIP, the World Climate Research Programme's Working Group on Regional Climate, and the Working Group on Coupled Modelling, former coordinating body of CORDEX and responsible panel for CMIP5. We thank the climate modelling groups for producing and making available their model output, the Earth System Grid Federation (ESGF) for archiving the data and providing access, and the multiple funding agencies who support CMIP5, CMIP6 and ESGF. We also acknowledge the Earth System Grid Federation infrastructure, an international effort led by the US Department of Energy's Program for Climate Model Diagnosis and Intercomparison, the European Network for Earth System Modelling and other partners in the Global Organisation for Earth System Science Portals (GO-ESSP). For CMIP the US Department of Energy's Program for Climate Model Diagnosis and Intercomparison provides coordinating support and led development of software infrastructure in partnership with the Global Organization for Earth System Science Portals.

This work has received funding from the European Union's Horizon 2020 research and innovation program under grant agreement No 820655 (EXHAUSTION) and from the Belmont Forum Collaborative Research Action on Climate, Environment, and Health, supported by the Research Council of Norway (contract No 310672, HEATCOST). Jana Sillmann and Anne Sophie Daloz were supported through the CICERO Strategic Project on Climate Change Risk (no. 160015/F40), funded by the Research Council of Norway.

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
