# Peer review of "High-resolution projections of ambient heat for major European cities using"

_Natural Hazards and Earth System Sciences, 2023_

## Author Comment (AC1)

We thank the reviewer for the valuable and constructive feedback on our manuscript. Below we explain point-by-point how we adjusted our manuscript based on the reviewer's suggestions.

Additionally, we have implemented the following more substantial modifications:

- **We added a table that summarizes the representation of urban areas in RCMs, and we elaborate on the different parameterizations and their potential impacts on our results in the discussion section.**
- **We added a new figure to the supplementary (Figure S2), which shows a map with the EURO-CORDEX biases relative to ERA5-Land and E-OBS in the 36 investigated European cities.**

The paper regards the evaluation of heat stress indicators for various cities in in Europe.

Quantification of heat stress under global warming is performed employing state-of-the-art GCMs and RCMs, and the authors correlates the various indicators to latitude, longitude, and proximity to the sea.

The statistical analysis is well performed, clearly presenting results and with robustness. From my point of view, the current work deserves a publication, after these minor points, referring in particular to 1) strengthen some sentence with reference to literature 2) Explaining more in detail the method employed for the statistical analysis:

LINE 32: ...such as in Canada in summer 2021 or in China and Europe in summer 2022. I would add a reference to this sentence, and/or a quantification through te registered anomaly

**We added two references for this sentence. Additionally, we changed summer 2022 to summer 2023 as the heat extremes in 2023 are even more recent, and a study by Zachariah et al. (2023) shows a clear relation between climate change and the 2023 heat extremes. The sentence now reads "[…] such as in Canada in summer 2021 (White et al., 2023) or in China and Europe in summer 2023 (Zachariah et al., 2023)." (lines 32-33)**

LINES 47-51. Here you present HWMId. However, there are more heat stress indicator, such as MRT or UTCI. I would extend the introduction about heat stress indicators, stressing why you employ HWMId.

**We extended the introduction part about the available heat metrics, additionally highlighting that various types of indicators exist and explaining, why we focus on temperature-based indicators:**

**"Various metrics have been developed with the aim to capture the characteristics of heat extremes, including heatwaves, and their potential evolution in the future (e.g., Perkins and Alexander, 2013; Perkins, 2015; de Freitas and Grigorieva, 2017). Several of these indicators are based solely on temperature, while others consider additional factors, such as humidity, solar radiation, or wind speed to estimate heat exposure (de Freitas and Grigorieva, 2017). In the following, we focus on temperature-based metrics, given that many epidemiological studies found temperature to be the dominant factor for adverse health effects (Armstrong et al., 2019; Kent et al., 2014; Vaneckova et al., 2011)." (lines 40-45)**

LINES 66-67: Analyses of climate and climate change in cities face the challenge of delivering results on spatial resolutions that are high enough to be relevant for cities while robustly estimating the risk of extreme events. I would add a reference to this sentence

**This sentence is intended as an opening for the discussion about the potentials and limitations of urban canopy layer models on the one side and regional and global climate models on the other side. The following sentences explain the potentials and limitations of both approaches more in detail and include the respective references. We thus decided to leave this opening sentence without further references, given that the references appear in the following sentences.**

LINES 67: "Urban models" is too general in your topic. I would call them "Urban canopy parameterizations"

**Thank you for pointing us to this ambiguity. Throughout the paper we changed "urban models" to "urban canopy layer models", following the formulation used by Goret et al. (2019).**

LINE 83: ...simulations at a resolution of 0.11° (EUR-11, ~12.5 km), which is fine enough to analyze climate conditions in major European cities at the city-level. In my opinion this resolution is too coarse. So to convince me, you should justify your sentence.

**We added "as typically at least one model grid cell falls within the extent of each major European city" (lines 87-88). To illustrate this, Figure 1 also shows the grid spacing of the EURO-CORDEX models compared to the extent of three European capitals of different size.**

LINES 144-145: we use the grid cell closest to the city centre for our analysis.

How do you define the city center? You should say it since you use it as the center even for the subsequent averaging over surrounding cells.

**In fact, it is not trivial to obtain a list of city center coordinates that is consistent for different cities. The exact coordinates vary depending on the data source, and the data sources do usually not provide reasons why the indicated coordinates were selected for the city centers. We searched different data portals and ultimately, we decided to use the city coordinates as they are indicated in Wikipedia (from https://en.wikipedia.org/wiki/List_of_European_cities_by_population_within_city_limits and from the main Wiki page of each of the remaining 36 European cities).**

**It is also noteworthy that despite the uncertain exact geographical definitions of city centers, the difference across datasets is typically very small (i.e., around 0.01°-0.02° resp. 1-2 km). Thus, the choice of the exact coordinates should not impact the findings of our study.**

METHODOLOGY in general: I would Add a subsection-paragraph saying if or which model employ a urban canopy parameterization. I know it is so relevant for your work, but since some of the models cited later could use a UCM, I would say which member use it, and which model with appropriate reference to literature.

**We added a table that summarizes the representation of urban areas in regional climate models (Table 1), accompanied by a short description of how RCMs represent urban areas (in section 2.1.2):**

**"The representation of urban areas varies considerably across RCMs (Table 1). Some RCMs represent urban areas as rock surfaces, others assume reduced vegetation and adjusted surface parameters (such as albedo and roughness) for urban areas, and one RCM even includes a sophisticated urban model." (lines 144-146)**

**Based on this information, we adjusted and extended the discussion of our results in light of the different urban parameterization implemented in the models:**

**"The ~12.5 km spatial resolution of the EUR-11 simulations enables a much more detailed assessment of climate variability and climate change at the city-level compared to GCMs, which have a much coarser spatial resolution (~100 km). Yet, most land surface modules of models in the 0.11° EURO-CORDEX ensemble only employ a simplified representation of urban areas (Table 1), which prevents the full exploitation of their high spatial resolution for studies focusing on urban areas. A few models represent urban areas as rock surfaces, thus neglecting the influence of urban vegetation on the surface energy balance and the influence of urban buildings on turbulence, radiation, and hydrology. Other models apply adjusted parameters (e.g., for albedo and roughness length) and a reduced vegetation cover in urban areas, and thus consider the characteristics of cities to some extent. One of the models uses a sophisticated urban land model, which includes various aspects of urban areas, such as urban canyons, different levels of urbanisation, and radiation and hydrology schemes specifically adapted for urban areas. Despite these substantial differences in how urban areas are represented, no direct link can be found between the general behaviour of the different models in the projection of ambient heat (e.g., comparatively high levels of ambient heat in HadREM3-GA7-05 and WRF381P, and comparatively low levels in HIRHAM5, RACMO22E, and COSMO-crCLIM-v1-1, with all of these models using the adjusted-parameter approach to represent urban areas) and their representation of urban areas (Figure 8, Table 1)." (lines 611-624)**

LINE 177: You should say what GSAT is (it appears the first time here)

**GSAT refers to "global mean surface air temperature". The acronym is explained in the previous sentence (in parentheses) (line 190)**

LINE 220: How is T calculated in your models? Is it a diagnostic 2m temperature, the temperature of the first layer of the model or what? You should clarify it.

**We specified this by replacing "temperature" with "near-surface air temperature" throughout the manuscript. Additionally, we added a sentence to highlight that this refers to 2 m temperature. "Near-surface air temperature refers to the temperature at 2 m height." (lines 139-140)**

LINE 229: However, we do not apply bias adjustment here due to the lack of reliable reference data, as urban areas are not specifically represented in the reference datasets ERA5-Land and E-OBS.

I don't get the point here. If E-OBS is observation, I guess cities are present, at least influencing the observation. Could you clarify this point?

**It is true that information about urban areas is included in the E-OBS dataset in case E-OBS uses data from weather stations within the city limits. This is mentioned in section 2.1.3: "For cities, in which temperature data from weather stations within the city limits are assimilated in ERA5-Land or considered in E-OBS, UHI might, however, be partly included." (lines 187-188)**

**As the number of weather stations varies considerably across the 36 investigated cities (see Figure 3), the robustness of bias adjustment will likely also vary across the cities. To prevent potential variations in the robustness of bias adjustment, we thus refrain from applying a standard bias adjustment technique. We state this now more clearly in the section about bias adjustment:**

**"However, we do not apply bias adjustment here due to the lack of reliable reference data, given that urban areas are not specifically represented in the reference datasets ERA5-Land, and E-OBS only implicitly includes information about urban areas to the extent weather stations are present within the city limits (which does not apply to all analysed cities, see Figure 3)." (lines 245-248)**

LINE 234: ...is calculated for each grid cell in a box of 5x5 grid cells around the centre of each city in the reference period 1981-2010. Ok, I think I don't agree with this methodology. In fact, cities taken into account in this work varies substantially in size, so I don't get why, for example, a "small" city such as Lisbon should cover more than 50x50 km. I would strengthen this methodology justifying why you use this method.

**We are aware that the cities that we analyze vary in size, and thus each city has different numbers of grid cells within its limits. The rationale for using 5x5 grid cells for the simple bias adjustment is, however, not to represent the city itself but to define an area around the center of each city that is representative for the larger-scale climatological conditions. Bias adjustment based on a single grid cell would be prone to high statistical uncertainty, whereas using 5x5 grid cells should deliver a more robust estimate of the typical climate conditions at the location of each city.**

**We extended the explanation for using the 5x5 grid cells in the following way: "The 5x5 box is used to represent the larger-scale climatological conditions within and around each city. The rationale is to reduce the statistical uncertainty by basing the adjustment on 25 grid cells instead of just one." (lines 254-255)**

LINE 304: Change "depending" with depends

**We replaced "temperatures change depending on the distance" by "temperatures vary as a function of the distance" (line 331)**

LINE 311: Here you present the validation comparing the model with observation.. Did you find any trend in the systematic error, for example depending on latitude/longitude/altitude/distance from sea? It can be really helpful tu interpret the results.

**To better visualize the spatial patterns of the biases, we added a new figure to the Supplementary Information (new Figures S2), where we show maps of the multi-model median EURO-CORDEX biases relative to ERA5-Land and E-OBS. We mention these spatial bias patterns now also in Section 3.1.**

**For TX: "In general, a negative-to-positive tendency from North to South can be identified for the EURO-CORDEX biases (Supplementary Figure S2)." (lines 318-319)**

**For TN: "[...], although the spatial patterns differ from the bias patterns of maximum temperatures (Supplementary Figure S2). Biases are highest in northern, eastern, and southern European cities, while they are lowest in central European cities." (lines 325-327)**

LINE 325: How do you choose wether a city is close to the sea or not? You should mention it somewhere in the paper.

**We added a sentence that specifies how we define it: "A city is considered to be located close to the sea if it is directly adjacent to the sea." (lines 130-131)**

LINES 338-345: HEre you say that there is a strong correlation between city and the metric employed, reporting some example. Can you justify someway those behaviours?

**We rephrased the paragraph to make our statements clearer, and we added potential reasons for the different behaviors in different cities:**

**"In several cities, all considered heat metrics show high levels of ambient heat under 3 °C European warming (e.g., Athens, Belgrade, Bucharest, Madrid, Milan, Sofia, Zagreb). For other cities, however, the ambient heat levels differ substantially depending on the metric under consideration. Barcelona, for example, ranks number one in terms of HWMId-TX, but exceeds 30 °C only rarely. Lisbon has substantial increases in TXx and temperatures often exceed 30 °C, but HWMId-TX is rather low. Kazan has substantial increases in TXx and high HWMId-TX values, but TX exceedances above 30 °C are relatively low. Oslo ranks among the cities with weakest changes in TXx and with lowest TX exceedances above 30 °C, but with high HWMId-TX values. These discrepancies may be due to several reasons. For instance, cities with comparatively cooler climate may see large increases in TXx and high HWMId-TX values without having substantial exceedances above 30 °C. Cities with high climatological variability in TXx may have comparatively low HWMId-TX values despite large increases in TXx and, vice versa, relatively low increases in TXx might result in high HWMId-TX values in case of low climatological variability in TXx. Considering only one heat metric might thus lead to unbalanced conclusions about projections of ambient heat for urban areas, potentially underestimating future risks from heat stress." (lines 364-375)**

**Additionally, we also added the potential reasons for the discrepancies to the discussion:**

**"In some cities, the ranking varies considerably depending on the considered heat metric (particularly in Barcelona, Oslo, Lisbon, Warsaw, and Berlin; Figure 4), indicating that the choice of metrics may strongly influence projections of ambient heat in these cities. These discrepancies in the ambient heat estimates from different heat metrics depend, for instance, on the local climate conditions, as the number of days exceeding 30 °C is strongly connected to the average summer temperatures in a city (see Figure 5a) and HWMId values are influenced by the local temperature variability (see Eq. (1))." (lines 580-584)**

LINE 450: ... box of 3x3 grid cells around the center. Here again the issue of 1) city center and 2) city dimension as noted in the previous comments

**Most of our results are based on a single grid cell, i.e., the one that is located closest to the city center. Choosing only one grid cell may, however, lead to high statistical uncertainty, particularly if the surrounding grid cells have very different land cover characteristics or in case of orographic**

gradients. **We thus perform a sensitivity analysis, in which we analyze how strongly the results change within the 3x3 grid cells around the grid cell that lies closest to the city center. As for the 5x5 box used for the simple bias adjustment, the 3x3 box is not meant to represent the city itself, but the purpose is to test how the results change if different grid cells around the city center are selected for the analysis.**

LINE 522:...showing an increase in heatwave risk in southern Europe along with substantial increases in coastal regions in northern Europe.

Could you explain why there is this substantial increase in particular in coastal regions in Northen Europe?

**We hypothesize that this is connected to the amplified warming of the Baltic Sea compared to the surrounding land areas as projected by the EURO-CORDEX models (see Supplementary Figure 8). We added the following sentence to the discussion:**

**"Supplementary Figure S8 also shows that the EURO-CORDEX models project an amplified warming of the Baltic Sea compared to the surrounding land areas, which is likely the reason for the high values of HWMId-TX in northern European coastal cities." (lines 560-562)**

LINE 579:...UHI is projected to only intensify gradually due to global warming...

I don't think it is true. Other papers say the opposite, like Tewari et al. 2019 (Interaction of urban heat islands and heat waves under current and future climate conditions and their mitigation using green and cool roofs in New York City and Phoenix, Arizona).

I would justify why those papers you cite say so, or I would introduce your statement in another way.

**Thank you for pointing us to this helpful paper. We have revised the paragraph about the future evolution of UHI, highlighting the importance of UHI and the additional analyses that would be possible if RCMs included more sophisticated representations of urban areas:**

**"In our analysis, we do not find pronounced UHI effects (Figure 3, Supplementary Figure S3), which is likely related to the simplified representation of urban areas in RCMs. UHI may additionally increase in the future due to global warming (Koomen and Diogo, 2017; Tewari et al., 2019) and urban expansion (Huang et al., 2019; Koomen and Diogo, 2017), and UHI can further be elevated during heatwaves (Ward et al., 2016). More sophisticated representations of urban areas in RCMs would make it possible to assess how the EURO-CORDEX models project future UHI developments, and could facilitate sensitivity studies to identify the contributions of climate change, local climate feedbacks, and urbanisation to the projected increase of ambient heat in cities." (lines 640-646)**

---

## Author Comment (AC2)

**We thank the reviewer for the valuable and constructive feedback on our manuscript. Below we explain point-by-point how we adjusted our manuscript based on the reviewer's suggestions.**

**Additionally, we have implemented the following more substantial modifications:**

- **We added a table that summarizes the representation of urban areas in RCMs, and we elaborate on the different parameterizations and their potential impacts on our results in the discussion section.**
- **We added a new figure to the supplementary (Figure S2), which shows a map with the EURO-CORDEX biases relative to ERA5-Land and E-OBS in the 36 investigated European cities.**

The paper presents a thorough analysis of ambient heat projections in major European cities using the EURO-CORDEX ensemble, comparing them with data from E-OBS, ERA5-Land, and weather stations. The study evaluates temperature biases, uncertainties, and factors influencing spatial patterns. It highlights variations in biases across cities and emphasizes the role of downscaling by regional climate models in shaping temperature estimates. The paper introduces a novel examination of nighttime ambient heat and compares projections with CMIP5 and CMIP6 ensembles. While the paper offers valuable insights, providing additional methodological details and discussing the limitations of the EURO-CORDEX data in the context of cities and urban areas could enhance its clarity and impact.

Urban Processes:

One noteworthy concern I have regarding the paper is the potential limitation arising from the EURO-CORDEX dataset's lack of representation of urban processes. The absence of urban-specific factors like the urban heat island (UHI) effect in the EURO-CORDEX models might lead to an incomplete understanding of how local urban conditions could influence temperature distributions. Addressing this limitation explicitly and discussing its potential implications for the reliability of the findings could benefit the readership.

**We added additional information about the representation of urban areas in RCMs, as a response to a comment from the other reviewer. Table 1 now summarizes the representation of urban areas in the regional climate models, accompanied by a short description of the table in section 2.1.2:**

**"The representation of urban areas varies considerably across RCMs (Table 1). Some RCMs represent urban areas as rock surfaces, others assume reduced vegetation and adjusted surface parameters (such as albedo and roughness) for urban areas, and one RCM even includes a sophisticated urban model." (lines 144-146)**

**Based on this information, we adjusted and extended the discussion of our results in light of the different urban parameterization implemented in the models:**

**"The ~12.5 km spatial resolution of the EUR-11 simulations enables a much more detailed assessment of climate variability and climate change at the city-level compared to GCMs, which have a much coarser spatial resolution (~100 km). Yet, most land surface modules of models in the 0.11° EURO-CORDEX ensemble only employ a simplified representation of urban areas (Table 1), which prevents the full exploitation of their high spatial resolution for studies focusing on urban areas. A few models represent urban areas as rock surfaces, thus neglecting the influence of urban vegetation on the surface energy balance and the influence of urban buildings on turbulence,**

**radiation, and hydrology. Other models apply adjusted parameters (e.g., for albedo and roughness length) and a reduced vegetation cover in urban areas, and thus consider the characteristics of cities to some extent. One of the models uses a sophisticated urban land model, which includes various aspects of urban areas, such as urban canyons, different levels of urbanisation, and radiation and hydrology schemes specifically adapted for urban areas. Despite these substantial differences in how urban areas are represented, no direct link can be found between the general behaviour of the different models in the projection of ambient heat (e.g., comparatively high levels of ambient heat in HadREM3-GA7-05 and WRF381P, and comparatively low levels in HIRHAM5, RACMO22E, and COSMO-crCLIM-v1-1, with all of these models using the adjusted-parameter approach to represent urban areas) and their representation of urban areas (Figure 8, Table 1)."**
**(lines 611-624)**

Clarity of Methodology:

While the section on "Identifying factors influencing spatial patterns" is intriguing, the exact statistical methods used to establish the relationships between climate and location factors and heat metrics and the limitations of these methods should be explicitly stated.

**Section 2.4.1 provides the description of the statistical methods used to calculate how much of the spatial variability of the different heat metrics can be explained by the climate factors and by the location factors, respectively. To make this clear, we added a reference to section 2.4.1 in the first sentence of section 3.3. In the same sentence, we now highlight better that we do not analyze the relationships between climate and location factors, but that we separately analyze how much of the spatial patterns of ambient heat can be related to climate factors and to location factors, respectively. The revised sentence now reads:**

**"To better understand the spatial patterns of ambient heat projected by the different heat metrics, we estimate how much of the spatial variance is explained 1) by different climate factors, representing each city's temperature climatology as well as its projected changes, and 2) by different location factors (Figure 5; see Section 2.4.1 for methodological details)." (lines 385-387)**

**The limitations of the applied method are stated in Section 2.4.1. One limitation are potential collinearities of the explanatory variables: "The explanatory variables (i.e., the climatological factors or the location factors) may be correlated, and their contributions cannot be strictly disentangled." (lines 285-286).**

**An estimate for the uncertainty introduced by the collinearity of different explanatory variables can be obtained from the variability of the squared semipartial correlation estimates. We use this variability as an uncertainty estimate for the contribution of each explanatory variable (see Section 2.4.1 and Figure 5). Additionally, we highlight now more in detail that the employed correlation analysis does not allow any statements about causality:**

**„The variability of the squared semipartial correlation estimates is a measure for collinearities between the explanatory variables and can be used as an uncertainty estimate for the contribution of each explanatory variable. The estimated contribution of each explanatory variable to the spatial variability of each heat metric does not permit statements about causality, as it is purely based on correlation analysis. Instead, the contributions should be interpreted as a measure of the extent to which the explained variables may be influenced by the location of each city or by the climatic conditions and climate change at the location of each city." (lines 293-298)**

Comparative Analysis:

The comparison between EURO-CORDEX, CMIP5, and CMIP6 ensembles in Section 3.5 is a valuable addition. However, the paper could provide more insights into the potential reasons behind the differences in projections. Elaborating on the distinct characteristics of the GCMs and RCMs, such as spatial resolution and physical parameterizations, could enhance the understanding of the results.

**To highlight the differences between RCMs and GCMs, we added a paragraph to section 2.1.2 where we introduce the climate model data:**

**"The GCMs and RCMs used in this study differ in several aspects. Most importantly, the RCMs have a much higher spatial resolution (~12.5 km) than the GCMs (~100 km), and orography and coastlines are thus represented much more accurately in RCMs. GCMs and RCMs also differ in their projections of atmospheric aerosols over the European domain, with GCMs using future scenarios with decreasing atmospheric aerosol concentrations while RCMs assume a constant atmospheric aerosol load (Boé et al., 2020; Gutiérrez et al., 2020; Nabat et al., 2020). Additionally, unlike GCMs, several RCMs do not consider plant physiological $CO_2$ effects, which might cause an underestimation of temperature extremes (Schwingshackl et al., 2019)." (lines 158-163)**

**Additionally, we added a paragraph to the discussion, where we elaborate on how the differences between GCMs and RCMs might impact our results.**

**"In many of the investigated cities, CMIP5 and CMIP6 project higher increases in TXx and larger HWMId-TX values than EURO-CORDEX. This is likely caused by discrepancies in external forcing data and differences in process implementation (see Section 2.1.2). Specifically, the CMIP5 and CMIP6 simulations are based on future scenarios with decreasing atmospheric aerosol concentrations over the European domain, while the EURO-CORDEX simulations assume a constant atmospheric aerosol load (Boé et al., 2020). The RCMs of EURO-CORDEX may thus underestimate future warming in Europe as they do not consider the amplified warming from the additional solar radiation reaching and heating the Earth's surface in Europe because of the decreasing aerosol concentrations. In addition, unlike CMIP5 and CMIP6 GCMs, several RCMs do not consider plant physiological effects (Schwingshackl et al., 2019). The closing of plant stomata due to higher $CO_2$ concentrations and the associated decrease in latent and increase in sensible heat fluxes, which lead to enhanced extreme temperatures, are thus not fully captured by RCMs. These differences between GCMs and RCMs suggest that RCMs likely underestimate future levels of ambient heat in European cities. Yet, for several southern European cities the EURO-CORDEX models project considerably more days exceeding 30 °C than CMIP5 and CMIP6. In coastal cities, such as Istanbul, Athens, and Lisbon, these differences are likely due to the higher spatial resolution of EURO-CORDEX, which enables a better distinction of land and ocean grid cells. In other cities, like Madrid or Rome, better resolved orography might be the reason for the more frequent exceedances in EURO-CORDEX. Yet the causes for some discrepancies remain unclear, for instance for the more frequent exceedances above 30 °C projected by EURO-CORDEX for Milan, which lies in the rather flat Po Valley, or for the coastal city Barcelona, where EURO-CORDEX shows much fewer exceedances above 30 °C than CMIP5 and CMIP6." (lines 563-579)**